# BSLoRA: Enhancing the Parameter Efficiency of LoRA with Intra-Layer and Inter-Layer Sharing

Yuhua Zhou [*1 2]  Ruifeng Li [*1 2]  Changhai Zhou [3]  Fei Yang [2]  Aimin Pan [2]

## Abstract

Low-Rank Adaptation (LoRA) is a widely adopted parameter-efficient fine-tuning method for large language models (LLMs) to adapt to downstream tasks. However, in scenarios where multiple LoRA models are deployed simultaneously, standard LoRA introduces substantial trainable parameters, resulting in significant memory overhead and inference latency, particularly when supporting thousands of downstream tasks on a single server. While existing methods reduce stored parameters via parameter sharing, they fail to capture both local and global information simultaneously. To address this issue, we propose the **B**i-**S**hare **LoRA** (**BSLoRA**), which extends local LoRA with intra-LoRA and inter-LoRA parameter sharing to better capture local and global information. This approach reduces trainable parameters while maintaining or even enhancing model performance. Additionally, we design three transformation methods to improve the compatibility and collaborative efficiency of shared parameters with varying shapes, enhancing overall adaptability. Experiments on the 7B, 8B, and 13B versions of Llama show that BSLoRA, with only 44.59% of the parameters of standard LoRA, outperforms LoRA by approximately 0.33% on commonsense reasoning and 2.08% on MMLU benchmarks. Code is available at https://github.com/yuhua-zhou/BSLoRA.git.

*Equal contribution [1]College of Computer Science and Technology, Zhejiang University [2]Zhejiang Lab [3]School of Computer Science, Fudan University. Correspondence to: Aimin Pan <panaimin@zhejianglab.org>, Fei Yang <yangf@zhejianglab.org>.

*Proceedings of the 42nd International Conference on Machine Learning*, Vancouver, Canada. PMLR 267, 2025. Copyright 2025 by the author(s).

## 1. Introduction

Large language models (LLMs), such as GPT-4o (Openai, 2023) and Claude-3 (Anthropic, 2024), have recently shown remarkable generalization capabilities across a wide range of natural language tasks (Raiaan et al., 2024; Chang et al., 2024; Zhang et al., 2023), primarily due to the increase in model parameters. For example, GPT-3 (Brown et al., 2020) has 175 billion parameters, while the largest version of Llama 3 (Touvron et al., 2023) reaches 405 billion. However, the ever-increasing size of these models presents significant challenges for fine-tuning, as full parameter fine-tuning becomes computationally expensive and memory-intensive. To address this issue, Parameter-Efficient Fine-Tuning (PEFT) methods have been introduced, achieving performance comparable to full fine-tuning by adjusting only a small subset of parameters (Han et al., 2024) while keeping the majority of the model parameters frozen. Among these methods, LoRA (Hu et al., 2022) stands out by approximating parameter updates using the product of two low-rank matrices and has gained increasing popularity.

However, as model parameters grow, LoRA fine-tuning of LLMs still introduces a large number of additional parameters, even with a low rank. For example, using a LoRA rank of 64 on Llama 70B adds approximately 360 million parameters (1.4GB of memory), increasing training difficulty, which worsens when multiple LoRA services are deployed simultaneously (Wang et al., 2024). Additionally, LoRA parameters consume significant memory during inference, which increases loading and task-switching latency. Therefore, reducing LoRA's trainable parameters has become an urgent necessity.

Several existing methods aim to reduce LoRA's parameters through parameter-sharing (Mao et al., 2024; Sun et al., 2022b), with inter-layer sharing proving effective in reducing redundancy across different layers. For example, VeRA (Kopiczko et al., 2024) and VB-LoRA (Li et al., 2024) leverage inter-layer sharing to reduce memory and computation costs while capturing global patterns. However, these methods often overlook the local information and intra-layer parameter redundancy (Lin et al., 2024). In Transformer models, for instance, the attention heads and feed-forward

networks in the same layer often process similar features, leading to redundant parameters. **Therefore, there is a need to design a new sharing technique that captures both local and global features, while ensuring the shared parameters can be adapted to all modules with different shapes.**

In this paper, we analyze LoRA parameters and identify a high degree of redundancy (Figure 1). Meanwhile, our design follows MultiLoRA (Wang et al., 2023), which demonstrates that multi-LoRA structure can yield better results. Preliminary studies (Table 5) further confirm that intra-layer and inter-layer parameter sharing can reduce parameter count while maintaining model performance and effectively capturing global information. Accordingly, we propose **B**i-**S**hare LoRA (BSLoRA), a Multi-LoRA architecture that extends LoRA parameters into multiple smaller LoRAs to enhance fine-tuning expressiveness, where two smaller LoRAs are shared within and across layers to reduce parameters. Specifically, we decompose the LoRA matrices into three components: local parameters, which capture module-specific information; intra-layer shared parameters, which are shared within the same layer to capture local consistent features; and inter-layer shared parameters, which are shared across layers to capture global patterns. This enables BSLoRA to learn both local and global information efficiently. Additionally, to tackle the challenge of adapting shared parameters to all modules with different shapes, we present three shape transformation methods: Slice Sharing, Gate Transformation, and Kronecker Extension. To validate the effectiveness of BSLoRA, we conduct extensive experiments on the Llama model family across multiple commonsense reasoning and MMLU benchmarks. Our results demonstrate that BSLoRA achieves significant parameter savings of about 50% while maintaining or even improving the model's performance compared to standard LoRA and other existing methods. We also conduct experiments to analyze the benefits of rank value and contributions of different configurations for local and shared weights. In summary, our contributions are as follows:

- We propose BSLoRA, a unified sharing method that combines local parameters with intra-layer and inter-layer shared parameters to effectively capture both local and global information. This approach significantly reduces the number of trainable parameters while maintaining performance.

- We introduce three shape transformation techniques to handle varying parameter shapes, thus increasing the flexibility and effectiveness of parameter sharing.

- We conduct extensive experiments on multiple tasks, demonstrating the effectiveness of BSLoRA in reducing parameter redundancy and improving parameter efficiency.

## 2. Background and Motivation

### 2.1. Low-Rank Adaptation

LoRA fine-tuning is employed to recover performance with minimal parameter updates. For an LLM consisting of $l$ layers, the weight matrix of each layer $\boldsymbol{W}$ is adjusted using an update matrix $\Delta \boldsymbol{W} \in \mathbb{R}^{m \times n}$. This matrix is factorized into two low-rank matrices, $\boldsymbol{A}$ and $\boldsymbol{B}$, where $\boldsymbol{A} \in \mathbb{R}^{r \times m}$ and $\boldsymbol{B} \in \mathbb{R}^{n \times r}$, with $r$ being a hyperparameter shared by all layers. The effectiveness of fine-tuning is highly dependent on rank selection. In this approach, the original weight matrix $\boldsymbol{W}$ remains frozen, while only $\Delta \boldsymbol{W}$, represented by the product $\boldsymbol{AB}$, is updated. The forward computation can be expressed as:

$$f(\boldsymbol{x}) = (\boldsymbol{W} + \Delta \boldsymbol{W})\boldsymbol{x} = \boldsymbol{W}\boldsymbol{x} + \boldsymbol{B}\boldsymbol{A}\boldsymbol{x} \ . \qquad (1)$$

Given that the rank $r$ is typically much smaller than the dimension $d$, the computational cost is significantly reduced from $d^2$ to $2dr$. This optimization can reduce the trainable parameters during the learning process. Typically, the matrix $\boldsymbol{A}$ is initialized by a Gaussian distribution with a small standard deviation, and $\boldsymbol{B}$ is initialized as a zero matrix. Hence, at the beginning of fine-tuning, the model behaves identically to the pre-trained model.

### 2.2. Motivation

In large language models (LLMs), parameter redundancy is a common issue. Redundancy commonly occurs within the same transformer block where the attention layer and MLP layer have overlapping functions or learn similar patterns (Lin et al., 2024), and the parameters across different blocks where similar feature representations might be learned in multiple layers (Li et al., 2024). We also plot the parameter similarity in Figure 1, which shows the high similarity across different modules (details refer to Appendix A.2). Existing methods aim to address the parameter redundancy through simple parameter sharing techniques (Kopiczko et al., 2024; Li et al., 2024; Song et al., 2024). However, our preliminary experiments (results shown in Table 5 (Individual)) indicate that simple intra-layer or inter-layer sharing may result in some performance degradation. It may need module-specific parameters to learn the local features. Therefore, we consider introducing local parameters combined with shared parameters to fine-tune the LLM.

Inspired by Wang et al. (2023) and Tian et al. (2024), which identified that decomposing a large LoRA weight into multiple LoRA modules in parallel can enhance the model's adaptability and flexibility. Therefore, we intend to decomposed the entire set of LoRA parameters into three smaller LoRA parameter blocks: the first part acts on individual modules, the second part is shared among modules within the same layer, and the third part is shared by all modules.

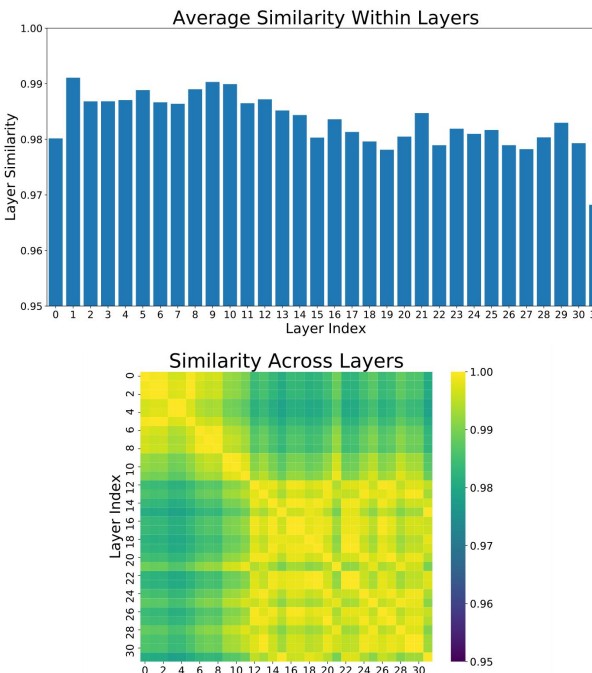

*Figure 1.* The entropy similarity (Lin et al., 2024) of LoRA parameters for each module within the same layer (top) and across different layers (bottom). It shows that different modules within the same layer exhibit high entropy similarity, and this high similarity is also present across different layers. This indicates that LoRA parameters have a significant degree of redundancy.

We assign different ranks for each smaller LoRA, the configuration and results are shown in Table 5 (Joint). The experimental results indicate that sharing parameters not only reduces the number of parameters but also improves model performance.

## 3. Method

### 3.1. Parameter Sharing

In this paper, we apply parameter sharing to enhance parameter efficiency in LoRA fine-tuning by leveraging the benefits of parameter sharing (Han et al., 2024). Our approach first decomposes the LoRA module horizontally, expanding the optimization search space and allowing for more efficient learning. We decompose the LoRA matrix into three sub-LoRA modules: **local**, **intra-layer**, and **inter-layer** which collectively contribute to the parameter updates of the target module.

**Intra-Layer Module**   refers to the sharing of parameters within the same transformer layer, such that all modules within a single layer (e.g., attention and MLP) share the same LoRA update matrix. This enables the model to capture consistent patterns and correlations within the layer,

thereby improving the coherence of the information processed within each layer.

**Inter-Layer Module**   denotes global parameter sharing, where the same LoRA update matrix is shared across different layers of the transformer. This facilitates better information flow and feature interaction between layers, which in turn enhance the overall expressiveness and depth of the model's representations.

**Local Module**   refers to the traditional LoRA configuration, where the LoRA update matrix is applied only to the current module, allowing the model to learn highly specific local features. During training and inference, the parameters for each module are updated according to the following equation:

$$f(\boldsymbol{x}) = \boldsymbol{W}\boldsymbol{x} + (\boldsymbol{B}\boldsymbol{A})_{local}\boldsymbol{x} + (\boldsymbol{B}\boldsymbol{A})_{intra}\boldsymbol{x} + (\boldsymbol{B}\boldsymbol{A})_{inter}\boldsymbol{x} \ , \quad (2)$$

where the distinct rank values of the aforementioned three sub-LoRA matrices are denoted as $r_{local}$, $r_{intra}$, and $r_{inter}$, respectively. Specifically, $\boldsymbol{B}_{local} \in \mathbb{R}^{n \times r_{local}}$, $\boldsymbol{A}_{local} \in \mathbb{R}^{r_{local} \times m}$, $\boldsymbol{B}_{intra} \in \mathbb{R}^{n \times r_{intra}}$, $\boldsymbol{A}_{intra} \in \mathbb{R}^{r_{intra} \times m}$, $\boldsymbol{B}_{inter} \in \mathbb{R}^{n \times r_{inter}}$, and $\boldsymbol{A}_{inter} \in \mathbb{R}^{r_{inter} \times m}$.

By introducing this novel decomposition, our BSLoRA fine-tuning approach is capable of learning both localized features and global interactions, providing a balance between task-specific adaptability and overall model robustness. Figure 2 presents an overview of our BSLoRA. By decomposing LoRA into multiple smaller LoRA modules, we can assign a higher rank to the shared parameters, allowing the final parameters to achieve a greater overall rank, as discussed further in Section 4.4.

### 3.2. Shape Transformation

We initially followed the LoRA setup (Hu et al., 2022) by applying LoRA parameters to the $q$ and $v$ modules in the attention layer, matching the shared parameter size to the $qv$ module, which yielded preliminary results (Table 5). Previous studies suggest that applying LoRA to more modules improves performance (Dettmers et al., 2023), so we extended it to the FFN layers. However, this caused a parameter shape mismatch. In the Transformer block (Vaswani et al., 2017), the $qkvo$ modules in Llama's attention layer have a consistent shape of $(4096, 4096)$. Meanwhile, the FFN's up- and down-projection modules have dimensions of $(4096, 11008)$ and $(11008, 4096)$, respectively, making it difficult to apply a single shared $\boldsymbol{A}\boldsymbol{B}$ parameters across these varying shapes. Similarly, in Llama3, the Grouped-Query Attention (GQA) changes the shape of $k$ module, further complicating the parameter sharing. To address this issue, we develop three transformation methods that adjust shared parameters to the size of target modules.

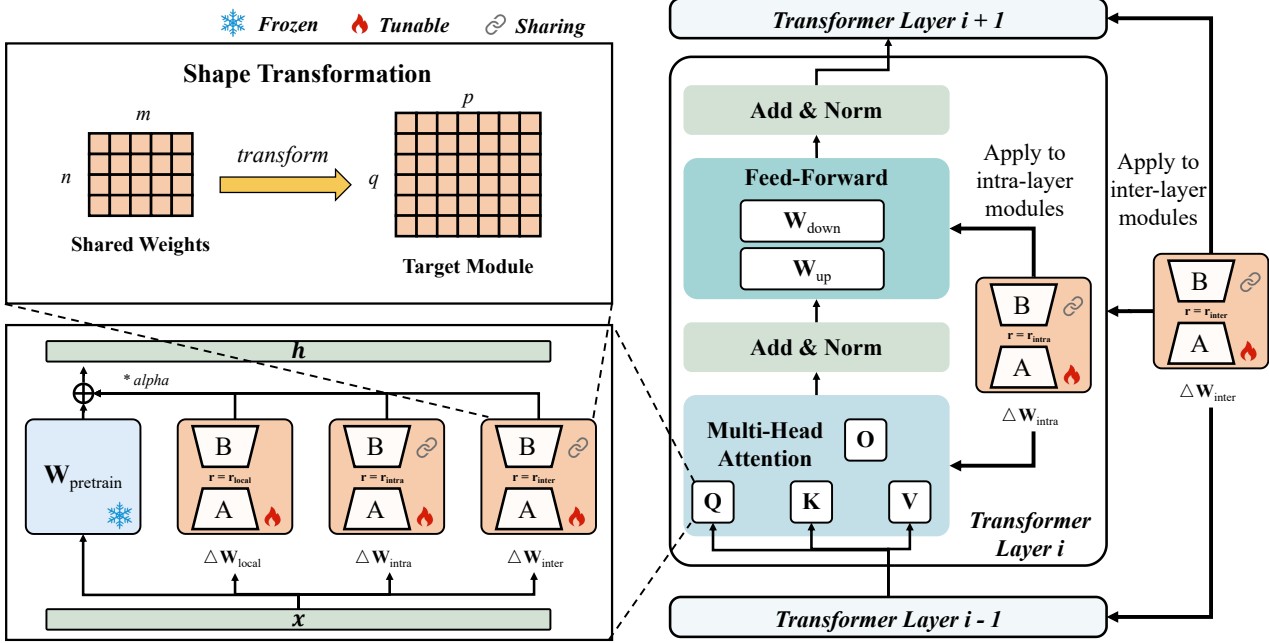

*Figure 2.* An overview of BSLoRA. The LoRA weights are decomposed into three sub-LoRA weights: *local* learns module-specific information, *intra* capture the shared features within the same transformer block, and *inter* learns the global information to interact with each module. The shape transformation enables the shared weights adaptive to different shapes of modules. By combining *local*, *intra*, and *inter*, the model can learn both local and global information during fine-tuning, so that improves the performance and generalizability.

### 3.2.1. SLICE SHARING

The straightforward method is to slice a larger trainable parameter matrix and train only the sliced portions (see Figure 3 (a)), which we refer to as the Slice Sharing (SS) method. Specifically, we determine the maximum input dimension $d_{im}$ and output dimensions $d_{om}$, among all fine-tuning parameter modules. The shared matrices are then defined with dimensions $\boldsymbol{A}_s \in \mathbb{R}^{r \times d_{im}}$ and $\boldsymbol{B}_s \in \mathbb{R}^{d_{om} \times r}$. During forward computation, the shared matrix is automatically sliced to match the parameter dimensions of the target module. The calculation is expressed as:

$$\Delta \boldsymbol{W} = \boldsymbol{B}_s[:, : d_o] \boldsymbol{A}_s[: d_i, :] \ , \tag{3}$$

where $d_i$ and $d_o$ represent the input and output dimensions of the target module, $\boldsymbol{B}_s[:, : d_o]$ and $\boldsymbol{A}_s[: d_i, :]$ represent the sliced parts of the shared weights of $\boldsymbol{B}_s$ and $\boldsymbol{A}_s$. Algorithm 1 provides the pseudocode for Slice Sharing.

### 3.2.2. GATE TRANSFORMATION

The simple slicing method enables parameter sharing, but only a subset of the shared parameters is used by all modules, while the remaining parameters are only utilized by larger modules. This limits the efficiency of parameter sharing. To address this, we propose matrix multiplication for dynamic size transformation of shared parameters. By multiplying matrices $\boldsymbol{M}_a \in \mathbb{R}^{m \times n}$ and $\boldsymbol{M}_b \in \mathbb{R}^{n \times p}$, the resulting matrix $\boldsymbol{M}_c \in \mathbb{R}^{m \times p}$ transforms the shape. Based on this,

we introduce the Gate Transformation (GT), which applies an input gate $\boldsymbol{G}_i \in \mathbb{R}^{m \times d_i}$ and an output gate $\boldsymbol{G}_o \in \mathbb{R}^{d_o \times n}$. For an input $\boldsymbol{x} \in \mathbb{R}^{b \times d_i}$, $\boldsymbol{G}_i$ transforms it to $(b, m)$, and the shared matrix $\boldsymbol{W}_s \in \mathbb{R}^{n \times m}$ processes it to produce an intermediate result $(b, n)$. Finally, $\boldsymbol{G}_o$ outputs the final shape $(b, d_o)$.

However, defining these transformation matrices introduces many learnable parameters, leading to high memory consumption for large inputs and outputs. To mitigate this, we apply one-rank decomposition to the input and output gates, reducing them to the product of two small rank-one matrices (see Figure 3 (b)). The final computation is as follows:

$$\Delta \boldsymbol{W} = \boldsymbol{G}_o \boldsymbol{W}_s \boldsymbol{G}_i = (\boldsymbol{G}_{ou} \boldsymbol{G}_{od})(\boldsymbol{B}_s \boldsymbol{A}_s)(\boldsymbol{G}_{iu} \boldsymbol{G}_{id}) \ , \tag{4}$$

where the $\boldsymbol{G}_{id} \in \mathbb{R}^{1 \times d_i}$ projects down the input dimension into a lower-dimensional space, and then $\boldsymbol{G}_{iu} \in \mathbb{R}^{m \times 1}$ scales it up into the dimension that is comparable to the input dimension of the shared weights $\boldsymbol{W}_s$. Similarly, $\boldsymbol{G}_{od} \in \mathbb{R}^{1 \times n}$ and $\boldsymbol{G}_{ou} \in \mathbb{R}^{d_o \times 1}$ are applied to transform the output. By setting the input and output gates, the size of our shared parameters can be flexibly changed. Algorithm 2 provides the pseudocode for Gate Transformation.

### 3.2.3. KRONECKER EXTENSION

While utilizing Gate Transformation allows us to define shared parameters of arbitrary shapes, low-rank decomposition may lead to information loss in both input and output

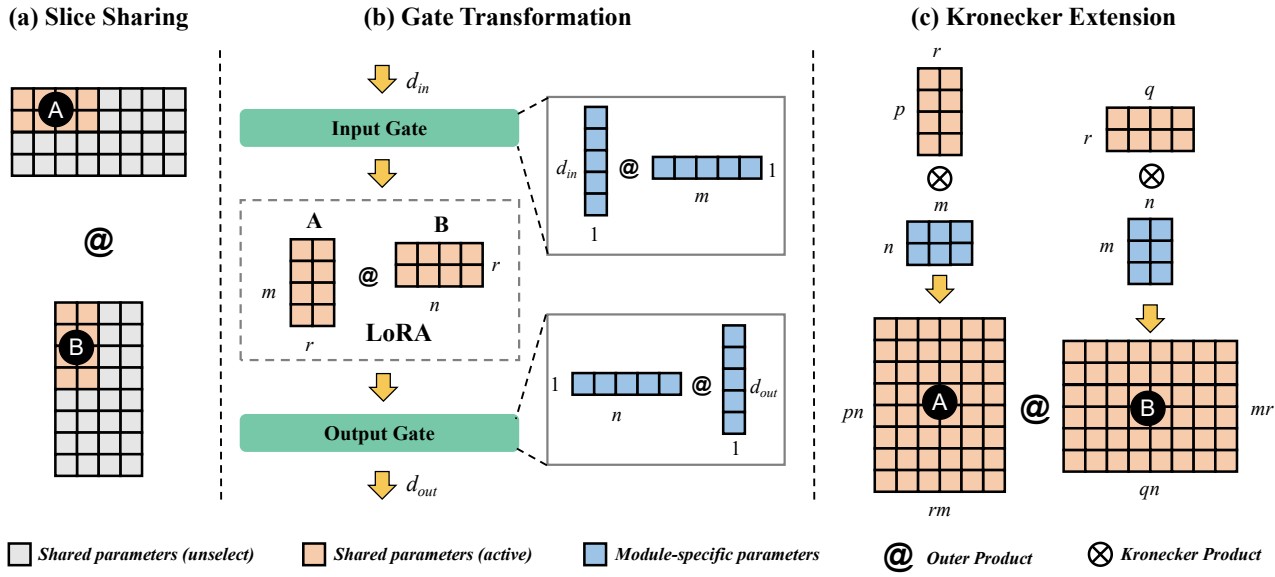

**(a) Slice Sharing**       **(b) Gate Transformation**       **(c) Kronecker Extension**

☐ *Shared parameters (unselect)*   ☐ *Shared parameters (active)*   ☐ *Module-specific parameters*   @ *Outer Product*   ⊗ *Kronecker Product*

*Figure 3.* Three methods for shape transformation: (a) Slice Sharing, slices parameters from a large shared trainable parameters; (b) Gate Transformation, an input gate and an output gate transforms input dimension and output dimension to match the shape of shared parameters, and they are implemented by one-rank decomposition; (c) Kronecker Extension, a module-specific kernel are used to extend the shared weights into target shape through the Kronecker Product.

transformations. Another approach is to concatenate multiple small shared parameters from identical copies to form a larger shared parameter (Wang et al., 2024; Edalati et al., 2022), but this limits the overall expressiveness. To address this, inspired by Karimi Mahabadi et al. (2021), we apply Kronecker matrix multiplication to expand the dimensions of the shared matrix by integer multiples, a method we term Kronecker Extension (KE). The Kronecker product between matrices $\boldsymbol{X} \in \mathbb{R}^{m \times n}$ and $\boldsymbol{Y} \in \mathbb{R}^{p \times q}$, denoted as $\boldsymbol{X} \otimes \boldsymbol{Y} \in \mathbb{R}^{mp \times nq}$, is mathematically defined as:

$$\boldsymbol{X} \otimes \boldsymbol{Y} = \begin{pmatrix} x_{11}\boldsymbol{Y} & \cdots & x_{1f}\boldsymbol{Y} \\ \vdots & \ddots & \vdots \\ x_{m1}\boldsymbol{Y} & \cdots & x_{mf}\boldsymbol{Y} \end{pmatrix} , \qquad (5)$$

where $x_{ij}$ shows the element in the $i^{th}$ row and $j^{th}$ column of $\boldsymbol{X}$.

We assign a module-specific kernel $\boldsymbol{K} \in \mathbb{R}^{d \times r}$ to each module in the model (see Figure 3(c)). By applying the Kronecker product, we expand the shared parameter $\boldsymbol{A}_s \in \mathbb{R}^{r \times \frac{m}{k}}$ and $\boldsymbol{B}_s \in \mathbb{R}^{\frac{n}{k} \times r}$ to match the size $\boldsymbol{A}_t \in \mathbb{R}^{r \times m}$ and $\boldsymbol{B}_t \in \mathbb{R}^{n \times r}$ of the target module. Here, $r$ is the rank value set for the shared parameter. Finally, according to our Kronecker Extension, $\Delta \boldsymbol{W}$ is calculated as:

$$\Delta \boldsymbol{W} = (\boldsymbol{K}_B \otimes \boldsymbol{B}_s)(\boldsymbol{K}_A \otimes \boldsymbol{A}_s) , \qquad (6)$$

where $\boldsymbol{K}_B \in \mathbb{R}^{k \times 1}$ and $\boldsymbol{K}_A \in \mathbb{R}^{1 \times k}$ represent module-specific kernels for the $\boldsymbol{B}$ and $\boldsymbol{A}$ matrices in the LoRA module. Algorithm 3 provides the pseudocode for Kronecker Extension.

## 4. Experiments

### 4.1. Settings

**LLMs.** To demonstrate how BSLoRA performed on different models, we conduct experiments on Llama families (Touvron et al., 2023): Llama 1 and Llama 3. In particular, we fine-tune the 7B and 13B models of Llama 1 and the 8B model of Llama 3, with the specific versions detailed in Appendix B.2.

**Benchmark.** We conduct experiments for these LLMs on two different benchmarks. The first benchmark is **Commonsense Reasoning**, which includes BoolQ (Clark et al., 2019), PIQA (Bisk et al., 2020), HellaSwag (Zellers et al., 2019), WinoGrande (Sakaguchi et al., 2021), ARC-easy (Clark et al., 2018), ARC-challenge (Clark et al., 2018), OpenbookQA (Mihaylov et al., 2018), and SIQA (Sap et al., 2019). The second benchmark is **Massively Multitask Language Understanding (MMLU)** (Hendrycks et al., 2021). Dataset details are presented in Appendix B.3. We employed lm-eval-harness (Gao et al., 2023) to create open prompts for the benchmarks and produce the results.

**Baselines.** We compare against several recently proposed LoRA-based PEFT methods: **(1) LoRA** (Hu et al., 2022), we set the rank to 8 for the standard LoRA to fine-tune the model. **(2) VeRA** (Kopiczko et al., 2024), we adopt the default setting where the rank is set to 64. **(3) VB-LoRA** (Li et al., 2024), we follow the setting of VB-LoRA, where the vector length is set to 256 and there are 90 vectors to be trained. Moreover, we set the k of the top-k to 2. **(4) Tied-LoRA** (Renduchintala et al., 2024), we follow the setting

*Table 1.* Results of Zero-shot performance on Llama 1-7B, Llama 3-8B, and Llama 1-13B in BSLoRA and baselines on Commonsense Reasoning benchmark. We report the number of trainable parameters (# params) and the corresponding ratio for each method.

| | Methods | # params | ratio | OBQA | ARC-c | HellaSwag | ARC-e | PIQA | WinoG. | BoolQ | SIQA | Avg. |
|---|---|---|---|---|---|---|---|---|---|---|---|---|
| Llama 1-7B | $\text{LoRA}_{r=8}$ | 14.02M | 0.21% | 44.80 | 47.10 | 77.35 | 76.47 | 80.25 | 69.77 | 77.98 | 48.21 | 65.24 |
| | $\text{VeRA}_{r=64}$ | 0.89M | 0.01% | 44.60 | 44.80 | 76.47 | 75.88 | 79.27 | 70.24 | 75.14 | 46.37 | 64.09 |
| | $\text{VB-LoRA}_{r=4}$ | 2.49M | 0.04% | 46.00 | 47.35 | 77.28 | 77.44 | 79.98 | 70.48 | 76.79 | 48.77 | 65.51 |
| | $\text{ShareLoRA}_{r=8}$ | 7.23M | 0.11% | 44.40 | 47.53 | 77.49 | 77.36 | 80.03 | 70.48 | 78.04 | 48.06 | 65.42 |
| | $\text{Tied-LoRA}_{r=8}$ | 0.44M | 0.01% | 44.40 | 44.88 | 76.19 | 75.25 | 79.22 | 70.09 | 75.06 | 45.70 | 63.85 |
| | BSLoRA (SS) | 7.03M | 0.10% | 46.20 | 46.93 | 77.23 | 76.77 | 80.52 | 69.85 | 78.13 | 49.39 | 65.63 |
| | BSLoRA (GT) | 8.22M | 0.12% | 45.20 | 47.27 | 77.47 | 77.06 | 80.14 | 70.17 | 78.93 | 48.93 | 65.64 |
| | BSLoRA (KE) | 3.66M | 0.05% | 45.40 | 47.87 | 77.32 | 77.57 | 80.20 | 70.24 | 77.52 | 48.62 | 65.59 |
| Llama 3-8B | $\text{LoRA}_{r=8}$ | 14.16M | 0.18% | 46.20 | 57.34 | 80.04 | 82.95 | 81.88 | 73.72 | 82.32 | 48.67 | 72.42 |
| | $\text{VeRA}_{r=64}$ | 0.80M | 0.01% | 45.00 | 54.01 | 79.27 | 80.51 | 81.23 | 73.32 | 81.07 | 47.34 | 70.96 |
| | $\text{VB-LoRA}_{r=4}$ | 2.51M | 0.03% | 46.40 | 56.06 | 79.85 | 81.27 | 81.39 | 74.51 | 81.62 | 46.93 | 71.66 |
| | $\text{ShareLoRA}_{r=8}$ | 8.06M | 0.11% | 45.60 | 56.57 | 79.89 | 82.79 | 81.99 | 73.80 | 82.69 | 48.46 | 72.31 |
| | $\text{Tied-LoRA}_{r=8}$ | 0.44M | 0.01% | 45.00 | 53.16 | 79.17 | 80.09 | 80.85 | 73.24 | 81.07 | 47.24 | 70.69 |
| | BSLoRA (SS) | 7.67M | 0.10% | 46.40 | 57.17 | 79.96 | 82.95 | 81.94 | 74.74 | 83.09 | 49.03 | 72.70 |
| | BSLoRA (GT) | 8.03M | 0.10% | 46.20 | 56.83 | 79.89 | 82.87 | 81.94 | 74.27 | 82.97 | 48.36 | 72.45 |
| | BSLoRA (KE) | 3.83M | 0.05% | 46.40 | 56.57 | 80.04 | 83.08 | 82.15 | 73.64 | 82.60 | 48.98 | 72.44 |
| Llama 1-13B | $\text{LoRA}_{r=8}$ | 21.95M | 0.17% | 45.40 | 51.71 | 80.21 | 79.21 | 80.90 | 72.69 | 81.13 | 48.87 | 67.52 |
| | $\text{VeRA}_{r=128}$ | 1.40M | 0.01% | 44.80 | 47.87 | 79.30 | 77.61 | 80.25 | 72.85 | 78.07 | 46.88 | 65.95 |
| | $\text{VB-LoRA}_{r=8}$ | 3.88M | 0.04% | 47.20 | 51.11 | 80.91 | 78.66 | 80.58 | 72.38 | 80.18 | 49.49 | 67.56 |
| | $\text{ShareLoRA}_{r=8}$ | 11.25M | 0.09% | 45.80 | 51.71 | 80.38 | 79.34 | 80.58 | 72.85 | 80.89 | 49.03 | 67.57 |
| | $\text{Tied-LoRA}_{r=8}$ | 0.55M | 0.01% | 44.80 | 47.87 | 79.08 | 77.40 | 80.25 | 72.85 | 77.92 | 56.65 | 65.85 |
| | BSLoRA (SS) | 10.13M | 0.08% | 45.80 | 51.28 | 80.11 | 79.21 | 80.74 | 72.69 | 81.59 | 49.13 | 67.57 |
| | BSLoRA (GT) | 12.14M | 0.9% | 46.00 | 51.02 | 80.11 | 79.04 | 80.90 | 72.53 | 80.95 | 48.72 | 67.41 |
| | BSLoRA (KE) | 5.94M | 0.05% | 45.00 | 51.79 | 80.32 | 79.21 | 80.74 | 72.85 | 80.83 | 48.93 | 67.46 |

of $TL_5$. **(5) ShareLoRA** (Song et al., 2024), we adopt the ShareA configuration to finetune the LLM.

**Fine-tuning Dataset.** We utilized publicly available samples from the Alpaca dataset (Taori et al., 2023) [1] to further fine-tune the LLM, which contains 52k instruction-following demonstrations generated by OpenAI's text-davinci-003 engine.

**Hyper-parameters and Training Details.** We apply the LoRA weights to the $W_q$, $W_k$, $W_v$, $W_{up}$, and $W_{down}$ modules of each Transformer block. For each shape transformation method, we set different rank configurations $r = \{r_{local}, r_{intra}, r_{inter}\}$. We set $r = \{2, 4, 32\}$ for Slice Share (SS). We set the shape of shared weights for Gate Transformation (GT) to $(1024, r_{share})$ and $r = \{2, 8, 16\}$. For the Kronecker Extension (KE), we set $r = \{2, 4, 16\}$ to adapt the shared weights' shape of $(256, r_{share})$. We use the same training configurations to fine-tune the LLM with BSLoRA and baseline methods. Specifically, we use AdamW (Loshchilov & Hutter, 2019) as the optimizer with 100 warm-up steps and a learning rate of $1 \times 10^{-4}$ and set the batch size to 64. For all the experiments, we train for one epoch.

[1] https://huggingface.co/datasets/yahma/alpaca-cleaned

### 4.2. Results on Commonsense Reasoning

We evaluate the zero-shot performance of BSLoRA on Commonsense Reasoning tasks using Llama 1-7B, Llama 3-8B, and Llama 1-13B models. In Table 1, the results show that BSLoRA consistently outperforms the baselines in terms of average performance across these datasets. Specifically, the Kronecker Extension (KE) method introduces fewer trainable parameters while achieving comparable performance to the SS and GT methods, indicating its superior parameter efficiency. More details in Appendix A.4.1.

### 4.3. Results on MMLU Benchmark

We evaluate the zero-shot and five-shot performance of BSLoRA on the MMLU benchmark using Llama 1-7B, Llama 3-8B, and Llama 1-13B models. The results, as shown in Table 3, demonstrate that BSLoRA consistently outperforms the baseline models in terms of average performance across both zero-shot and five-shot settings. This highlights the effectiveness of our approach in diverse scenarios (see more in Appendix A.4.2).

### 4.4. Analysis

**Rank Analysis.** According to matrix rank theory, the rank of the sum of two matrices is given by $\mathcal{R}(A+B) \leq \mathcal{R}(A) + \mathcal{R}(B)$. This implies that decomposing a large LoRA matrix into multiple sub-LoRAs does not increase the overall rank

*Table 2.* Results of zero-shot and five-shot performance on Llama 1-7B, Llama 3-8B, and Llama 1-13B in BSLoRA and baselines on MMLU benchmark. We report the number of trainable parameters (# params) and the corresponding ratio for each method.

| | Method | # params | ratio | MMLU (0-shot) | | | | | MMLU (5-shot) | | | | |
|---|---|---|---|---|---|---|---|---|---|---|---|---|---|
| | | | | Hums. | STEM | Social | Other | Avg. | Hums. | STEM | Social | Other | Avg. |
| Llama 1-7B | LoRA$_{r=8}$ | 14.02M | 0.21% | 34.67 | 31.24 | 37.21 | 39.36 | 35.62 | 34.86 | 32.57 | 40.36 | 40.81 | 37.15 |
| | VeRA$_{r=64}$ | 0.89M | 0.01% | 32.22 | 28.32 | 32.40 | 36.72 | 32.42 | 33.58 | 31.27 | 38.58 | 38.85 | 35.57 |
| | VB-LoRA$_{r=4}$ | 2.49M | 0.04% | 30.10 | 28.51 | 29.25 | 33.31 | 30.29 | 34.11 | 30.54 | 40.14 | 40.52 | 36.33 |
| | ShareLoRA$_{r=8}$ | 7.32M | 0.11% | 33.01 | 30.61 | 33.57 | 37.62 | 33.70 | 33.88 | 30.73 | 38.93 | 39.78 | 35.83 |
| | Tied-LoRA$_{r=8}$ | 0.44M | 0.01% | 31.92 | 28.20 | 32.11 | 36.31 | 32.13 | 33.84 | 31.56 | 38.64 | 38.53 | 35.64 |
| | BSLoRA (SS) | 7.03M | 0.10% | **36.34** | **32.98** | 40.59 | **42.55** | **38.12** | **36.62** | **33.46** | **42.48** | **43.35** | **38.98** |
| | BSLoRA (GT) | 8.22M | 0.12% | 36.30 | 32.92 | **40.75** | 42.13 | 38.03 | 35.15 | 32.19 | 41.44 | 41.94 | 37.68 |
| | BSLoRA (KE) | 3.66M | 0.05% | 35.56 | 32.00 | 38.12 | 40.01 | 36.42 | 35.32 | 32.41 | 41.40 | 41.29 | 37.61 |
| Llama 3-8B | LoRA$_{r=8}$ | 14.16M | 0.18% | 56.77 | 53.89 | 72.77 | 70.36 | 63.44 | 59.81 | 55.92 | 76.21 | 72.10 | 66.01 |
| | VeRA$_{r=64}$ | 0.80M | 0.01% | 54.88 | 54.17 | **73.35** | 70.58 | 63.25 | **59.85** | 55.69 | 76.15 | **72.80** | 66.12 |
| | VB-LoRA$_{r=4}$ | 2.51M | 0.03% | 55.83 | 52.74 | 71.86 | 71.00 | 62.86 | 59.11 | 55.22 | 74.68 | 72.26 | 65.32 |
| | ShareLoRA$_{r=8}$ | 7.67M | 0.11% | 56.81 | 54.17 | 73.16 | 70.87 | 63.75 | 59.68 | 55.63 | 76.18 | 72.61 | 66.02 |
| | Tied-LoRA$_{r=8}$ | 0.44M | 0.01% | 54.77 | 54.01 | 74.29 | 70.58 | 64.16 | 59.96 | 55.82 | 76.05 | 72.64 | 66.12 |
| | BSLoRA (SS) | 7.67M | 0.10% | **58.53** | **54.61** | 73.35 | 71.29 | **64.45** | 59.57 | **56.26** | 75.95 | 72.35 | 66.04 |
| | BSLoRA (GT) | 8.03M | 0.10% | 58.30 | 54.39 | 73.29 | **71.48** | 64.37 | 59.68 | 56.23 | **76.31** | 72.48 | **66.18** |
| | BSLoRA (KE) | 3.83M | 0.05% | 57.98 | 54.49 | 73.35 | 70.84 | 64.16 | 59.38 | 55.98 | 76.08 | 72.16 | 65.90 |
| Llama 1-13B | LoRA$_{r=8}$ | 21.95M | 0.17% | 43.66 | 38.31 | **54.44** | 53.33 | 47.43 | 45.06 | 37.46 | **55.64** | 55.42 | 48.39 |
| | VeRA$_{r=128}$ | 1.40M | 0.01% | 41.66 | 36.50 | 48.75 | 48.73 | 43.91 | 44.23 | 37.08 | 53.92 | 53.59 | 47.20 |
| | VB-LoRA$_{r=8}$ | 3.88M | 0.04% | 41.02 | 35.81 | 49.66 | 49.76 | 44.06 | 44.27 | 37.14 | 54.27 | 53.91 | 47.40 |
| | ShareLoRA$_{r=8}$ | 11.25M | 0.09% | 43.78 | 37.33 | 52.75 | 52.78 | 46.77 | 44.61 | 36.63 | 54.01 | 54.23 | 47.37 |
| | Tied-LoRA$_{r=8}$ | 0.55M | 0.01% | 41.72 | 36.38 | 48.52 | 48.44 | 43.76 | 44.46 | 37.01 | 53.85 | 53.59 | 47.23 |
| | BSLoRA (SS) | 10.13M | 0.08% | **44.76** | 37.46 | 53.04 | 53.72 | 47.24 | 44.78 | 37.81 | 54.50 | **55.65** | 48.18 |
| | BSLoRA (GT) | 12.19M | 0.09% | 43.61 | 37.77 | 53.33 | 53.14 | 46.96 | 44.46 | 37.52 | 53.88 | 55.07 | 47.73 |
| | BSLoRA (KE) | 5.94M | 0.05% | 44.48 | **38.98** | 54.34 | **54.49** | **48.07** | **45.29** | **37.84** | 55.25 | 55.36 | **48.43** |

of the matrix. Consequently, the rank of the combined LoRA matrices remains bounded by the sum of their individual ranks. We validate the actual rank of each module and calculate the average ranks across layers for each Shape Transformation method using rank configurations of $r = \{2, 4, 16\}$. The results, shown in Figure 4(a), indicate that the Slice Sharing (SS) method achieves a combined rank equal to the sum of local and shared ranks (22), while the Kronecker Extension (KE) reaches approximately 21.53. In contrast, the Gate Transformation (GT) method yields rank value equivalent to the local rank plus 2, likely due to one-rank gates causing some information loss.

**Contribution Analysis.** We conduct further experiments to explore the contribution of each sub-LoRA matrix. By setting the rank of one sub-LoRA matrix to 8 and the others to 0, we examine the individual impact of each component. The Kronecker Extension method is used to reshape the shared parameters. In Figure 4(b), the results reveal the performance preferences of each component across datasets. Specifically, the local component of LoRA performs best on HellaSwag, ARC-e, BoolQ, and SIQA, while the intra-layer shared component excels on PIQA and ARC-c. Overall, the combination of local, intra-layer, and inter-layer shared parameters yields the best performance across all datasets.

**Extension Analysis.** We further analyze the scalability of the Kronecker Extension method. The shared matrix

$W \in \mathbb{R}^{n \times r}$ can be obtained by applying the Kronecker product to $M \in \mathbb{R}^{\frac{n}{k} \times r}$ and $K \in \mathbb{R}^{k \times 1}$. By adjusting their shapes to $M \in \mathbb{R}^{\frac{n}{k} \times 1}$ and $K \in \mathbb{R}^{k \times r}$, the same matrix shape for $W$ can be achieved. Furthermore, since one dimension is a constant 1, the resulting matrix rank equals $r$. To explore its effect, we modify this constant to 2 to test whether it enhances the rank of $W$. We denote configurations as $a\_b$, where $r\_1$ means the shared matrix has a dimension $r$ and a module-specific dimension of 1, while $2\_r$ has a shared dimension of 2 and a module-specific dimension of $r$. The results in Figure 4(c) show that modifying the constant improves both expressiveness and information content. Comparing $r\_1$ and $1\_r$, we observe that each configuration excels in different metrics, with $r\_1$ performing better overall while introducing fewer trainable parameters.

### 4.5. Ablation Study

We conduct ablation studies to examine the impact of individual components of our method. All subsequent experiments focus on the MMLU (zero-shot) (Hendrycks et al., 2021) and GSM8K (5-shot) (Cobbe et al., 2021) benchmarks, utilizing the Llama 1-7B model. We also conduct experiments to examine the impact of different rank allocations in Appendix A.4.5.

**Shared Size of Gate and Kronecker Kernel.** The Gate Transformation (GT) and Kronecker Extension (KE) meth-

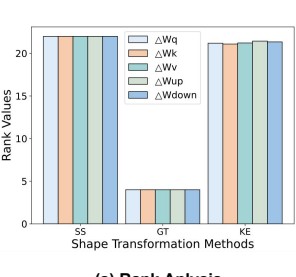
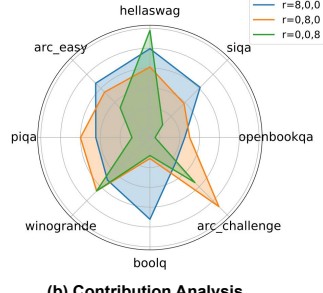
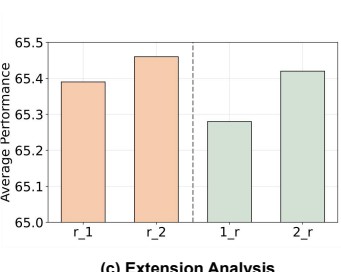

| (a) Rank Anlysis | (b) Contribution Analysis | (c) Extension Analysis |

*Figure 4.* Analysis. (a) **Rank Analysis**: the final rank benefit from different Shape Transformation methods. (b) **Contribution Analysis**: the different contribution across the sharing weights and local weights. (c) **Extension Analysis**: the rank extension analysis for different shape of share weights' size and kernel size.

ods can flexibly adapt the shared weights to arbitrary shapes for different modules. We examine various sizes of the shared matrix, setting the rank $r = \{2, 8, 16\}$. For the GT, we test shared matrix sizes of [512, 1024, 2048], while for the KE, we test sizes of [64, 128, 256]. The results are presented in Table 3 and Table 4. We found that the parameters introduced by GT increase with the growth of the shared parameter size, which also leads to an improvement in performance on the MMLU benchmark. However, we observe a decrease in performance on the GSM8K dataset as the shared parameter size increases. In contrast, the parameters introduced by the KE method decrease with increasing shared parameter size, while the average performance improves. Overall, we select the shared parameter size of 1024 for the GT method and 256 for the KE method.

*Table 3.* Size of Shared weights of GT

| Shared Size | # params | raito | MMLU | GSM8K |
|---|---|---|---|---|
| 512 | 7.62M | 0.11% | **36.03** | 10.54 |
| 1,024 | 8.22M | 0.12% | 35.61 | 10.77 |
| 2,048 | 9.44M | 0.13% | 34.70 | **11.90** |

*Table 4.* Size of Shared weights of KE

| Shared Size | # params | raito | MMLU | GSM8K |
|---|---|---|---|---|
| 64 | 4.07M | 0.06% | 36.47 | 10.84 |
| 128 | 3.82M | 0.05% | 35.34 | **11.98** |
| 256 | 3.72M | 0.05% | **37.01** | 11.22 |

## 5. Related Work

### 5.1. Parameter Sharing of LoRA

Recent advances in LoRA-based fine-tuning methods have explored various strategies for sharing LoRA weights to enhance parameter efficiency across multiple tasks. VeRA (Kopiczko et al., 2024) proposes sharing random matrices across all layers, reducing the number of parameters, but it results in some performance trade-offs and increased inference latency due to its high-rank requirements. In addition, Tied-LoRA (Renduchintala et al., 2024) takes a different approach by sharing LoRA matrices specifically across query, key, and value projection layers, using additional scaling

vectors to differentiate the modules. However, its requirement for identical matrix shapes (appliable for q, k, and v matrices) limits flexibility. ShareLoRA (Song et al., 2024) devises to share A, B, or AB matrices across layers to reduce parameters, but it is only applicable for the weights with the same shape. In contrast, PRoLoRA (Wang et al., 2024) employs an intra-layer sharing mechanism with learnable parameters, but it only reduces parameters without capturing global features. Additionally, VB-LoRA (Li et al., 2024) introduces a "divide-and-share" approach that partitions shared vectors into a vector bank, addressing the limitations of traditional low-rank decomposition across matrix dimensions, modules, and layers. However, this approach selects vectors uniformly without accounting for the internal structure of the model, leading to suboptimal utilization of the model's internal information.

### 5.2. Multi-LoRA Architecture

LoRA has demonstrated exceptional resource efficiency and performance in adapting LLMs for specific tasks, driving the demand for a single model capable of handling multiple tasks (Agiza et al., 2024). Several approaches have been proposed to improve their multi-task adaptability. In particular, LoraHub (Huang et al., 2023) assembles LoRA modules trained on different tasks to eliminate the need for human expertise and assumptions, enabling effective cross-task generalization. Similarly, MultiLoRA (Wang et al., 2023) improves adaptability by horizontally expanding LoRA modules and reducing the dominance of top singular vectors observed in LoRA. Building on these advancements, HydraLoRA (Tian et al., 2024) introduces an asymmetric architecture that shares a common matrix across tasks while using task-specific matrices for different sub-domains, further enhancing both fine-tuning and inference efficiency.

## 6. Discussion

Recent work on S-LoRA (Sheng et al., 2024) addresses the deployment side of LoRA by keeping a unified GPU

buffer for adapter weights, dynamically paging seldom-used adapters from CPU to GPU, and fusing heterogeneous requests into a single batched GEMM. Although orthogonal in scope, S-LoRA and BSLoRA are highly complementary: **(1) Reduced paging overhead.** BSLoRA removes up to 56% of the trainable weights through hierarchical sharing. With fewer parameters per adapter, S-LoRA has to move considerably less data across the CPU-GPU boundary, which shortens task-switch latency and allows many more adapters to remain resident on chip. **(2) Shape-aware batching.** BSLoRA organises its shared components by transformation type—Slice, Gate, or Kronecker. Because each type yields identical weight shapes across tasks, S-LoRA's scheduler can co-group requests that reuse the same shared tensor and issue a single large GEMM, further improving arithmetic intensity. **(3) Persistent shared cores.** In BSLoRA the intra-layer and inter-layer matrices are tiny. Under S-LoRA these small cores can remain permanently pinned in GPU memory, while only the larger local blocks are paged to and from CPU. Consequently, the amount of data transferred per context switch is minimised, further reducing end-to-end latency. We believe a unified BSLoRA + S-LoRA serving stack will offer both model-level and system-level efficiency, enabling the simultaneous deployment of thousands of personalised adapters on a single GPU. Implementing and benchmarking this joint design is left for future work.

## 7. Conclusion

This paper presents BSLoRA, which enhances the parameter efficiency of LLMs by combining shared intra-layer, inter-layer parameters, and local parameters. This approach reduces the number of trainable parameters while boosting model efficiency. Experiments on various Llama models demonstrate that BSLoRA significantly cuts down parameter usage by 56.40% and improves average performance by 0.33% on commonsense reasoning tasks and 2.08% on MMLU benchmark. BSLoRA reduces redundancy, enhances model adaptability, and offers flexible sharing strategies with potential high-rank benefits.

## Acknowledgements

This project is based upon work supported by National Natural Science Foundation of China Grant No. U22A6001, National Key Research and Development Program of China No. 2023YFE0108600, Pioneer" and "Leading Goose" R&D Program of Zhejiang No.2024SSYS0002.

## Impact Statement

This paper presents work whose goal is to advance the field of Machine Learning. There are many potential societal consequences of our work, none of which we feel must be specifically highlighted here.

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

# A. Appendix

### A.1. Parameter-Efficient Fine-Tuning (PEFT)

Parameter-Efficient Fine-Tuning (PEFT) is crucial for large language models (LLMs) as it reduces computational costs while preserving performance (Ding et al., 2023). Specifically, Adapters (Houlsby et al., 2019; Zhu et al., 2025), inserted between layers, allow task-specific fine-tuning with minimal trainable parameters while keeping most weights fixed. Similarly, Prefix Tuning (Li & Liang, 2021) adds learnable tokens at each transformer layer to guide task-specific behavior without modifying core parameters. In contrast, Prompt Tuning (Lester et al., 2021) optimizes a small set of prompts attached to the input, leaving the model's architecture unchanged. LoRA (Hu et al., 2022) reparameterizes weight matrices into low-rank forms, significantly reducing trainable parameters while maintaining performance. Some studies build upon LoRA to further improve and optimize its performance (Shi et al., 2024; Ren et al., 2024; Jiang et al., 2024; Liu et al., 2024; Koohpayegani et al., 2024). Likewise, we propose BSLoRA, building on LoRA, which shares parameters both within and across layers to further enhance efficiency and adaptability.

### A.2. Parameter Similarity

#### A.2.1. ENTROPY QUANTIFICATION

Information entropy is a key concept in information theory, used to quantify the uncertainty or randomness within a dataset or signal. It provides a measure of the average information content per symbol or event in a message or sequence of messages. As defined by Guan et al. (2019), entropy can be employed to assess the information capacity of a network. Consequently, the entropy of a given layer can be calculated based on the probability distribution of its features:

$$H(F) = -\int p(f) \log p(f) \, \mathrm{df}, f \in F \ . \tag{7}$$

Nonetheless, it is difficult to directly measure the probability distribution of a feature map: $p(f), f \in F$. Following (Sirignano & Spiliopoulos, 2020; Sun et al., 2022a), we use the Gaussian distribution as the probability distribution of the intermediate feature in a layer. Therefore, the entropy of a certain layer is approximated as the mathematical expectation of $F \sim \mathcal{N}(\mu, \sigma^2)$:

$$\begin{aligned}
H(F) &= -\mathbb{E}\left[\log \mathcal{N}\left(\mu, \sigma^2\right)\right] \\
&= -\mathbb{E}\left[\log\left[\left(2\pi\sigma^2\right)^{-1/2} \exp\left(-\frac{1}{2\sigma^2}(f - \mu)^2\right)\right]\right] \\
&= \log(\sigma) + \frac{1}{2}\log(2\pi) + \frac{1}{2} \ ,
\end{aligned} \tag{8}$$

where $\sigma$ is the standard deviation of the feature set $f \in F$.

#### A.2.2. ENTROPY SIMILARITY

The information content of LoRA weight matrices can be measured by their entropy. Two matrices with similar entropy values typically contain similar amounts of information, indicating a high degree of redundancy. We assess the correlation between different weight matrices by calculating mutual information, which quantifies the relationship between them and can be computed using the following formula:

$$I(\boldsymbol{X}; \boldsymbol{Y}) = H(\boldsymbol{X}) + H(\boldsymbol{Y}) - H(\boldsymbol{X}, \boldsymbol{Y}) \ , \tag{9}$$

Here, $H(\boldsymbol{X}, \boldsymbol{Y})$ represents the entropy of the joint distribution of matrix $\boldsymbol{X}$ and matrix $\boldsymbol{Y}$. To compute this, we flatten the matrices $\boldsymbol{X}$ and $\boldsymbol{Y}$, concatenate them, and then calculate the entropy of the resulting joint distribution. A large mutual information between two weight matrices indicates significant overlap in their information, implying redundancy. The calculated mutual information $I(\boldsymbol{X}; \boldsymbol{Y})$ is an absolute value, which needs to be converted into a relative value. Therefore, we use Relative Mutual Information (RMI) to represent the similarity between matrices, calculated as follows:

$$RMI = \frac{I(\boldsymbol{X}; \boldsymbol{Y})}{\min(H(\boldsymbol{X}), H(\boldsymbol{Y}))} \tag{10}$$

If $RMI > 0.8$, the mutual information is considered high, indicating significant redundancy between the two matrices. If $0.5 < RMI \leq 0.8$, the mutual information is moderate, suggesting notable shared information but with some degree of independence. If $RMI \leq 0.5$, the mutual information is low, indicating minimal redundancy between the two matrices.

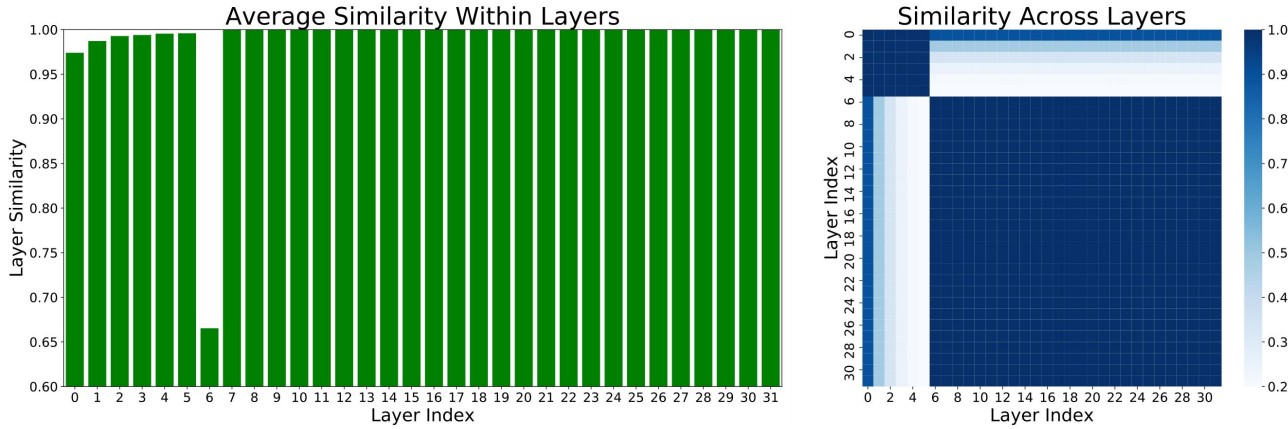

*Figure 5.* The entropy similarity of LoRA parameters for each module within the same layer (left) and across different layers (right). It shows that different modules within the same layer exhibit high entropy similarity, and this high similarity is also present across different layers. This indicates that LoRA parameters have a significant degree of redundancy.

We fine-tuned the Llama-7B model using LoRA with a rank $r = 64$, isolating the LoRA matrices and calculating $\Delta W$. The information entropy of $\Delta W$ for each module was computed based on Equation 8, and the Relative Mutual Information (RMI) between modules was calculated as a measure of similarity using Equations 9 and 10. We obtained the RMI values both within the same layer and across different layers, with the results shown in Figure 1. We observed that different modules within the same layer exhibit high entropy similarity, and this similarity also extends across layers. Additionally, following the method in Lin et al. (2024), we calculated the entropy similarity of the activation values for each module. As illustrated in Figure 5, the $\Delta W$ of different parameter modules also shows high similarity in terms of activation values, further indicating the significant redundancy in the parameters used in LoRA fine-tuning.

*Table 5.* Performance on instruction tuning with Alpaca 50K (Taori et al., 2023), evaluated with MMLU (Hendrycks et al., 2021) and GSM8K (Cobbe et al., 2021). **Boldface** indicates the best performance.

| Method | $r_{local}$ | $r_{intra}$ | $r_{inter}$ | # params | ratio | MMLU | GSM8K |
|---|---|---|---|---|---|---|---|
| Individual | | | | | | | |
| LoRA (no-share) | 8 | - | - | 4.19M | 0.06% | **35.12** | **10.84** |
| LoRA (intra-share) | - | 8 | - | 2.10M | 0.03% | 34.23 | 10.54 |
| LoRA (inter-share) | - | - | 8 | 0.07M | 0.01% | 32.20 | 10.08 |
| Joint | | | | | | | |
| LoRA (share-intra) | 4 | 4 | - | 3.14M | 0.05% | 34.63 | **10.92** |
| LoRA (share-inter) | 4 | - | 4 | 2.13M | 0.03% | 35.00 | 10.54 |
| LoRA (share-intra-inter) | 4 | 2 | 2 | 2.64M | 0.04% | **35.89** | 10.08 |

## A.3. Shape Transformation Algorithms

We provide detailed pseudocode for each of the Shape Transformation methods used in BSLoRA. Specifically, Algorithm 1 outlines the steps for the Slice Sharing method, which slices larger matrices to adapt to different parameter dimensions. Algorithm 2 demonstrates the Gate Transformation method, which utilizes input and output gates to dynamically adjust shared parameters for varying module shapes. Finally, Algorithm 3 explains the Kronecker Extension method, which expands shared matrices using Kronecker products to maintain consistency across modules with diverse dimensions. These algorithms collectively contribute to enhancing parameter efficiency and flexibility in model fine-tuning.

---

**Algorithm 1** Pseudocode of Slice Sharing.

---

**Input:** The rank of shared weights $r$ and the input $\boldsymbol{x}$;
**Output:** The output calculated by shared weights;
  1: *# Initialize the Shared weights:*
  2:  mdin, mdout $\leftarrow$ Find max InFeatures and OutFeatures across all modules;
  3:  $\boldsymbol{A} \in \mathbb{R}^{r \times mdin} \leftarrow$ Normal Randomly Initialization;
  4:  $\boldsymbol{B} \in \mathbb{R}^{mdout \times r} \leftarrow$ Zero Initialization;
  5:
  6: *# Training and Inference Stage:*
  7:  din, dout $\leftarrow$ InFeatures and OutFeatures of current module;
  8:  $\Delta \boldsymbol{W} \leftarrow \boldsymbol{B}[: dout, :]\boldsymbol{A}[:, : din]$
  9:  result $\leftarrow \Delta \boldsymbol{W} \boldsymbol{x}$
 10:
 11: **return** result

---

### A.4. Experiments

#### A.4.1. RESULTS ON COMMONSENSE REASONING

We evaluate BSLoRA for zero-shot performance on Commonsense Reasoning tasks using Llama 1-7B, Llama 1-13B, and Llama 3-8B. The results are shown in Table 1. We observe that BSLoRA consistently outperforms the baselines in terms of the average performance of Commonsense Reasoning datasets. Specifically, the Gate Transformation (GT) method of BSLoRA achieves the best average performance on the Llama 1-7B, while the Slice Sharing (SS) method achieves the best average performance on the Llama 3-8B. Furthermore, BSLoRA outperforms the baselines on 7 out of 8 datasets on Llama 1-7B, and 5 out of 8 datasets on Llama 3-8B. It is worth noting that large improvements are achieved on ARC-e, PIQA, BoolQ, and SIQA datasets. BSLoRA also achieves decent performance on the remaining datasets, including OBQA, HellaSwag, and WinG, which proves that BSLoRA is stable and reliable across different datasets. Compared to the standard LoRA with a rank of 8, BSLoRA can save about 50% trainable parameters and achieve better performance. Specifically, the Kronecker Extension (KE) method introduces fewer trainable parameters, meanwhile outperforms the baselines, and achieves on-par performance with the SS and GT methods, indicating the more parameter-efficient method.

#### A.4.2. RESULTS ON MMLU BENCHMARK

We evaluate BSLoRA for zero-shot and five-shot performance of MMLU tasks based on Llama 1-7B, Llama 1-13B, and Llama 3-8B. We demonstrate the results in Table 2. From the results, we observe that BSLoRA consistently outperforms the baselines in terms of the average performance of MMLU datasets both on zero-shot and five-shot. Specifically, the Slice Sharing (SS) method of BSLoRA achieves the best average performance on the Llama 1-7B, while the Gate Transformation (GT) method achieves the best average performance on the Llama 3-8B. Notice that the SS method on Llama 1-7B achieves the best performance on 9 out of 10 metrics and achieves the second-best performance for the remaining one. Furthermore, the GT and Kronecker Extension (KE) BSLoRA achieve the second-best performance compared to SS and consistently outperform the baselines. Compared to the standard LoRA with a rank of 8, BSLoRA can save about 50% trainable parameters and achieve better performance. Specifically, the KE method outperforms the baselines and achieves on-par performance with the SS and GT methods, indicating the most parameter-efficient method.

#### A.4.3. EXPERIMENTS ON FLAN_V2 DATASET

To further demonstrate the effectiveness of BSLoRA, we also conduct experiments on Llama 3-8B on the FLAN_V2 instruction dataset (Chung et al., 2022) [2], which is another dataset for instruction tuning. We conduct experiments on the Chain Of Thought task and evaluate its performance on the Commonsense Reasoning task datasets; the experiment results are shown in Table 6.

---

[2] https://huggingface.co/datasets/BEE-spoke-data/flan-v2-hf

---

**Algorithm 2** Pseudocode of Gate Transformation.

---

**Input:** The rank of shared weights $r$ and the input $x$;
**Output:** The output calculated by shared Weights;

1: *# Initialize the Shared Weights:*
2: dins, douts ← InFeatures and OutFeatures of shared weights;
3: $A_s \in \mathbb{R}^{r \times dins}$ ← Normal Randomly Initialization;
4: $B_s \in \mathbb{R}^{douts \times r}$ ← Zero Initialization;
5:
6: *# Initialize Gate weights:*
7: din, dout ← InFeatures and OutFeatures of current module;
8: $G_{id} \in \mathbb{R}^{1 \times din}$ ← Uniform Randomly Initialization;
9: $G_{iu} \in \mathbb{R}^{dins \times 1}$ ← Uniform Randomly Initialization;
10: $G_{od} \in \mathbb{R}^{1 \times douts}$ ← Uniform Randomly Initialization;
11: $G_{ou} \in \mathbb{R}^{dout \times 1}$ ← Uniform Randomly Initialization;
12:
13: *# Training and Inference Stage:*
14: $G_i \in \mathbb{R}^{dins \times din} \leftarrow G_{iu} G_{id}$;
15: $G_o \in \mathbb{R}^{dout \times douts} \leftarrow G_{ou} G_{od}$;
16: $\Delta W \leftarrow G_o B_s A_s G_i$
17: result ← $\Delta W x$
18:
19: **return** result

---

*Table 6.* Results of Zero-shot performance on Llama 3-8B, in BSLoRA and baselines on Commonsense Reasoning benchmark. We report the number of trainable parameters (# params) and the corresponding ratio for each method.

| | Methods | # params | ratio | OBQA | ARC-c | HellaSwag | ARC-e | PIQA | WinoG. | BoolQ | SIQA | Avg. |
|---|---|---|---|---|---|---|---|---|---|---|---|---|
| Llama 3-8B | LoRA$_{r=8}$ | 14.16M | 0.18% | 45.40 | 53.41 | 79.20 | 80.43 | 79.82 | 74.35 | 83.27 | 47.39 | 67.91 |
| | VeRA$_{r=64}$ | 0.80M | 0.01% | 44.80 | 53.92 | 79.14 | 79.92 | 79.54 | 72.69 | 80.95 | 47.03 | 67.25 |
| | VB-LoRA$_{r=4}$ | 2.51M | 0.03% | 44.00 | 54.01 | 78.72 | 80.26 | 78.62 | 74.66 | 81.19 | 46.01 | 67.18 |
| | ShareLoRA$_{r=8}$ | 8.006M | 0.11% | 44.80 | 54.27 | 79.04 | 80.89 | 81.23 | 74.19 | 83.00 | 48.57 | 68.25 |
| | Tied-LoRA$_{r=8}$ | 0.44M | 0.01% | 45.00 | 53.07 | 79.11 | 80.01 | 80.79 | 73.88 | 81.25 | 47.03 | 67.52 |
| | BSLoRA (SS) | 7.67M | 0.10% | **46.60** | **56.23** | 79.00 | 81.14 | 80.79 | 74.27 | 83.52 | 48.77 | 68.79 |
| | BSLoRA (GT) | 8.03M | 0.10% | 45.20 | 55.38 | **79.22** | 81.99 | **81.28** | **74.98** | 83.21 | 48.52 | 68.72 |
| | BSLoRA (KE) | 3.83M | 0.05% | 44.40 | 55.03 | 79.11 | **82.41** | 81.18 | 74.90 | **84.10** | **49.54** | **68.83** |

### A.4.4. ABLATION STUDY

**Slicing Method.** We evaluate three slicing methods for the shared matrix: top-left slice, bottom-right slice, and center slice. We adopt the rank configuration of $r = \{2, 4, 8\}$. The results are shown in Table 7. It indicates that the center slice outperforms the other two slicing methods both on MMLU and GSM8K benchmark.

**Gate Initialization.** We initialize the input and output gates of the Gate Transformation using three schemes: Kaiming normal, Kaiming uniform, and constant one initialization. The rank configurations for different sub-LoRA weights are set for $r = \{2, 8, 16\}$. As shown in Table 8, Kaiming uniform outperforms Kaiming normal initialization. Additionally, the constant ones initialization leads to gradient explosion or vanishing issues, making it unsuitable for gate initialization.

**Initialization of Kronecker Kernel.** We apply Kaiming normal, Kaiming uniform, and constant ones initialization to the Kronecker kernel to examine the impact of different initialization schemes. We set the ranks of $\{2, 4, 16\}$ for local, intra, and inter sub-LoRA matrices, respectively. From Table 9, the constant ones initialization performs better on the MMLU benchmark, while the Kaiming normal initialization outperforms the other two methods on the GSM8K benchmark. Overall, the Kaiming normal initialization performs best.

---

**Algorithm 3** Pseudocode of Kronecker Extension.

---

**Input:** The rank of shared weights $r$ and the input $\boldsymbol{x}$;
**Output:** The output calculated by shared weights;

 1: *# Initialize the Shared Weights:*
 2: dins, douts ← InFeatures and OutFeatures of shared weights;
 3: $\boldsymbol{A}_s \in \mathbb{R}^{r \times dins}$ ← Normal Randomly Initialization;
 4: $\boldsymbol{B}_s \in \mathbb{R}^{douts \times r}$ ← Zero Initialization;
 5:
 6: *# Initialize Kernel weights:*
 7: din, dout ← InFeatures and OutFeatures of current module;
 8: $k_a$ ← dins // r;
 9: $k_b$ ← douts // r;
10: $\boldsymbol{K}_A \in \mathbb{R}^{1 \times k_a}$ ← Uniform Randomly Initialization;
11: $\boldsymbol{K}_B \in \mathbb{R}^{k_b \times 1}$ ← Uniform Randomly Initialization;
12:
13: *# Training and Inference Stage:*
14: $\boldsymbol{A} \in \mathbb{R}^{r \times d_i} \leftarrow \boldsymbol{K}_A \otimes \boldsymbol{A}_s$;
15: $\boldsymbol{B} \in \mathbb{R}^{d_o \times r} \leftarrow \boldsymbol{K}_B \otimes \boldsymbol{B}_s$;
16: $\Delta \boldsymbol{W} \leftarrow \boldsymbol{B}\boldsymbol{A}$
17: result ← $\Delta W \boldsymbol{x}$
18:
19: **return** result

---

|  | Table 7. Split Position | |
|---|---|---|
| Method | MMLU | GSM8K |
| Top-Left | 36.27 | 10.69 |
| Right-Down | 35.84 | 10.92 |
| Center | **36.58** | **11.14** |

|  | Table 8. Gate Initialization | |
|---|---|---|
| Matrix Init. | MMLU | GSM8K |
| Kaiming Unif. | **36.36** | **11.90** |
| Kaiming Norm. | 36.09 | 10.31 |
| Ones | NAN | NAN |

|  | Table 9. Kernel Initialization | |
|---|---|---|
| Matrix Init. | MMLU | GSM8K |
| Kaiming Unif. | 35.34 | 11.14 |
| Kaiming Norm. | 35.15 | **12.05** |
| Ones | **35.84** | 11.14 |

### A.4.5. RANK ALLOCATION

We designed comparative experiments to explore the impact of adjusting different rank ratios on fine-tuning performance. Specifically, we varied the rank value of one sub-LoRA module while keeping the ranks of the other two fixed. We conducted experiments on the rank configurations for local, intra-sharing, and inter-sharing. The results are shown in Table 10. Assigning a lower rank to the local component and a higher rank to the shared parameters yielded better performance, further illustrating the redundancy in the standard LoRA parameters.

### A.5. Analysis

#### A.5.1. PARAMETER COUNT ANALYSIS

In this section, we compare the number of parameters of BSLoRA with the standard LoRA. Considering an LLM with $L$ layers, where each layer contains $M$ modules with hidden dimension $d$, the number of trainable parameters is equal to the model size (i.e., $LMd^2$) in full fine-tuning. LoRA reduces this number to $2LMdr$, where $r$ is the rank of two low-rank decomposed matrices. In BSLoRA, the trainable parameters consist of two parts: local parameters $\mathcal{L}$, computed as $2LMdr_{local}$, and shared parameters, which include intra-parameter sharing $\mathcal{S}_{intra}$ and inter-parameter sharing $\mathcal{S}_{inter}$. Therefore, the stored parameters can be represented by a triplet $\Theta = \{\mathcal{L}, \mathcal{S}_{intra}, \mathcal{S}_{inter}\}$. Note that the different Shape Transformation methods would result in different parameters of $\mathcal{S}_{intra}$ and $\mathcal{S}_{inter}$.

**Slice Sharing.** The parameters of $\mathcal{S}$ in Slice Sharing are shared across all modules. Specifically, the parameters of intra-sharing $\mathcal{S}_{intra} = 2Ldr_{intra}$, and the parameters of inter-sharing $\mathcal{S}_{intra} = 2dr_{inter}$.

*Table 10.* Different configurations of ranks for our BSLoRA (KE). **Boldface** denotes the best results in terms of the corresponding metrics, and underline means the second-best performance.

| Method | Ranks | OBQA | ARC-c | HellaSwag | ARC-e | PIQA | WinoG. | BoolQ | SIQA | Avg. |
|---|---|---|---|---|---|---|---|---|---|---|
| Adjust local | 2,4,16 | **45.00** | **47.78** | **77.45** | 76.85 | 79.92 | **69.69** | 78.07 | 48.57 | 65.42 |
| | 4,4,16 | 44.80 | 47.61 | 77.35 | 77.02 | **80.30** | 69.50 | 77.74 | 48.31 | 65.33 |
| | 8,4,16 | 44.80 | 47.70 | 77.33 | **77.19** | **80.30** | 69.53 | 77.80 | **48.82** | **65.43** |
| | 16,4,16 | 44.60 | 47.18 | 77.38 | 76.68 | 80.20 | **69.69** | **78.16** | 48.52 | 65.30 |
| Adjust intra | 2,2,16 | **45.40** | 46.93 | 77.43 | 76.56 | 80.14 | 69.77 | 78.20 | **48.93** | **65.42** |
| | 2,4,16 | 45.00 | **47.78** | **77.45** | 76.85 | 79.92 | 69.69 | 78.07 | 48.57 | **65.42** |
| | 2,8,16 | 44.80 | 47.44 | 77.36 | **77.06** | 79.43 | **70.24** | 77.43 | 48.52 | 65.29 |
| | 2,16,16 | 45.20 | 46.84 | 77.38 | 76.52 | **80.41** | 69.61 | **78.38** | 48.77 | 65.39 |
| Adjust inter | 2,4,8 | **45.40** | 47.01 | 77.32 | 76.73 | 80.09 | 70.17 | **78.17** | 48.41 | 65.41 |
| | 2,4,16 | 45.00 | **47.78** | **77.45** | 76.85 | 79.92 | 69.69 | 78.07 | **48.57** | **65.42** |
| | 2,4,32 | 44.20 | 47.35 | 77.28 | **76.94** | 79.92 | **70.48** | 77.86 | 48.31 | 65.29 |
| | 2,4,64 | 44.80 | 46.67 | 77.31 | 76.89 | **80.30** | 69.93 | 77.77 | 48.36 | 65.26 |

**Gate Transformation.** Different modules within and between layers share the parameters with the hidden dimension of $d_s$. Besides, each module contains an input gate and an output gate to transform the dimensions. Specifically, the parameters of intra-sharing $\mathcal{S}_{intra} = 2(Ld_s r_{intra} + M(d + d_s)))$, and the parameters of inter-sharing $\mathcal{S}_{intra} = 2(d_s r_{inter} + ML(d + d_s))$.

**Kronecker Extension.** Different modules within and between layers share the parameters with the hidden dimension of $d_s$. Besides, each module contains a Kronecker kernel $K \in \mathbb{R}^{1 \times k}$ to transform the dimensions. Specifically, the parameters of intra-sharing $\mathcal{S}_{intra} = 2L(d_s r_{intra} + Mk)$, and the parameters of inter-sharing $\mathcal{S}_{intra} = 2(d_s r_{inter} + 2MLk)$.

### A.5.2. MULTI-LORA SERVING ANALYSIS

In a multi-LoRA deployment, tasks share a common pre-trained model, adapted to each task by loading specific LoRA parameters. Typically, all LoRA parameters are preloaded into GPU memory to minimize task-switching latency and maximize GPU utilization. However, with many tasks, only frequently used parameters remain in GPU memory, while others are stored on the CPU, causing delays during frequent task switching. To mitigate this, we reduce the parameter size through sharing, allowing more parameters to fit in GPU memory.

We conducted a comparative experiment to analyze the GPU memory footprint of deploying different numbers of downstream tasks using the Llama model. Specifically, we deployed a Llama 1-7B base model on an V100 32G GPU and measured the memory usage for 100 to 1100 LoRA parameters and BSLoRA

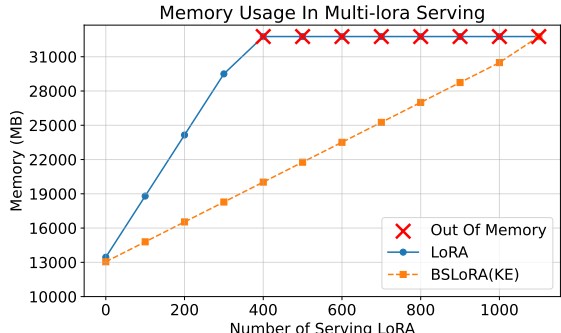

*Figure 6.* Memory usage comparison of LoRA and BSLoRA (KE) for serving different numbers of tasks.

(kE) parameters during inference. The results, shown in Figure 6, indicate that BSLoRA reduces memory usage by approximately 60% compared to standard LoRA under the same number of parameters. Furthermore, when deploying more than 350 LoRA parameters, the memory usage exceeds the GPU's capacity. Furthermore, to determine the maximum deployment capacity, we incrementally added parameters until the memory limit was reached. Our findings reveal that standard LoRA can load up to 341 parameters, while BSLoRA supports up to 1095, significantly enhancing deployment efficiency.

### A.5.3. CONTRIBUTION ANALYSIS

We further investigate how performance is affected when intra-layer and inter-layer parameter sharing are applied independently under different rank settings. Using the Llama 1-7B model fine-tuned on the Alpaca dataset, we configured various

shared rank settings for different shape transformation modalities and evaluated the results on the CommonSense task. The findings, presented in Table 11, demonstrate that intra-layer sharing alone outperforms inter-layer sharing alone in terms of performance. However, inter-layer sharing is significantly more parameter-efficient, achieving comparable results with fewer parameters. This trade-off underscores the complementary benefits of combining both strategies.

Additionally, we observed that using the GT approach to increase the rank for either intra-layer or inter-layer sharing alone does not lead to significant performance improvements and may even degrade performance. In contrast, for the SS and KE methods, increasing the rank consistently enhances model performance, further emphasizing the benefits of tailored sharing strategies.

*Table 11.* Different configurations of ranks for our BSLoRA. **Boldface** denotes the best results in terms of the corresponding metrics, and underline means the second-best performance.

| Method | Mode | Ranks | OBQA | ARC-c | HellaSwag | ARC-e | PIQA | WinoG. | BoolQ | SIQA | Avg. |
|---|---|---|---|---|---|---|---|---|---|---|---|
| | SS | 0,8,0 | 45.00 | **47.70** | **77.32** | 76.64 | **80.25** | 70.01 | 78.10 | 48.41 | 65.43 |
| | SS | 0,16,0 | 45.20 | 47.10 | 77.31 | **77.02** | 80.14 | **70.09** | 78.01 | 48.82 | 65.46 |
| | SS | 0,32,0 | **45.40** | 47.35 | 77.18 | 76.47 | 80.09 | 69.69 | **78.50** | **49.18** | **65.48** |
| | GT | 0,8,0 | 43.20 | **45.99** | **76.52** | 75.63 | 79.11 | 69.85 | **74.98** | **46.16** | **63.93** |
| Adjust intra | GT | 0,16,0 | **43.80** | 45.48 | 75.71 | 75.55 | **79.38** | 69.53 | 74.16 | 45.70 | 63.66 |
| | GT | 0,32,0 | 43.60 | 44.88 | 76.03 | 75.34 | 79.00 | **70.01** | 73.00 | 46.11 | 63.50 |
| | KE | 0,8,0 | 45.40 | **48.04** | 77.16 | 76.35 | **80.20** | **70.72** | 75.90 | 46.78 | 65.07 |
| | KE | 0,16,0 | **45.80** | 46.93 | 77.48 | **76.73** | 80.03 | 70.56 | 75.93 | **47.44** | **65.11** |
| | KE | 0,32,0 | 45.20 | 47.10 | **77.53** | 76.39 | 79.82 | 70.24 | **75.96** | 47.13 | 64.92 |
| | SS | 0,0,8 | **44.60** | 48.29 | 77.23 | **77.02** | **80.20** | 69.85 | 77.74 | **48.31** | 65.40 |
| | SS | 0,0,16 | 44.40 | **48.72** | 77.35 | 76.64 | 80.09 | 70.32 | 78.23 | 47.95 | 65.46 |
| | SS | 0,0,32 | 44.40 | 47.78 | **77.36** | 76.94 | 80.03 | **70.80** | **78.75** | 48.21 | **65.53** |
| | GT | 0,0,8 | **44.40** | 45.56 | 76.41 | 75.67 | **79.38** | **69.85** | 75.84 | 45.75 | 64.11 |
| Adjust inter | GT | 0,0,16 | 44.00 | **46.33** | **76.87** | **76.14** | 79.11 | 69.30 | 74.40 | **46.93** | **64.13** |
| | GT | 0,0,32 | **44.40** | 44.97 | 76.00 | 72.73 | 79.11 | 69.38 | 75.11 | 45.75 | 63.43 |
| | KE | 0,0,8 | 43.80 | 45.65 | 75.70 | 75.63 | **79.38** | 70.01 | 75.66 | 46.47 | 64.04 |
| | KE | 0,0,16 | 45.80 | **46.93** | **76.59** | **76.77** | 79.16 | **70.64** | 76.57 | 47.75 | **65.03** |
| | KE | 0,0,32 | **46.00** | 45.56 | 76.17 | 76.30 | 79.16 | 70.01 | **76.79** | **47.80** | 64.72 |

### A.5.4. BSLoRA Placement Analysis

To ensure a fair evaluation of parameter sharing contributions across module types, we conducted a controlled ablation study comparing three configurations: (1) applying BSLoRA exclusively to attention modules, (2) restricting BSLoRA to MLP modules, and (3) jointly applying BSLoRA to both attention and MLP modules.

Our experiments results (shown in Table 12) reveal two critical findings. First, BSLoRA achieves significantly higher performance when applied to attention modules alone compared to MLP-only adaptation, suggesting that attention layers exhibit more pronounced parameter redundancy. Second, the combined application of BSLoRA to both attention and MLP modules yields superior results over attention-only tuning, implying that redundancy exists not only across layers (inter-layer) but also across module types within individual layers (intra-layer).

## B. Future Work

In the future, we will explore more intelligent parameter sharing modes that can selectively select different modules or different layers for parameter sharing, ultimately further improving the performance of the model.

### B.1. LLM versions.

We provide the Hugging Face link of LLMs used in the experiment: Llama 1-7B: `https://huggingface.co/baffo32/decapoda-research-llama-7B-hf`; Llama 3-8B: `https://huggingface.co/meta-llama/`

*Table 12.* Different placement of modules for BSLoRA. **Boldface** denotes the best results in terms of the corresponding metrics, and underline means the second-best performance.

| Shared Modules | Methods | OBQA | ARC-c | HellaSwag | ARC-e | PIQA | WinoG. | BoolQ | SIQA | Avg. | Overall |
|---|---|---|---|---|---|---|---|---|---|---|---|
| Attention | BSLoRA (SS) | 46.00 | 56.48 | 83.09 | 82.91 | 81.83 | 73.72 | 83.09 | 48.00 | 68.39 | |
| | BSLoRA (GT) | 46.00 | 57.34 | 79.78 | 83.04 | 81.99 | 73.80 | 81.38 | 48.62 | 68.99 | 69.09 |
| | BSLoRA (KE) | 45.80 | 56.74 | 79.90 | 82.53 | 81.83 | 74.03 | 81.56 | 48.72 | 68.89 | |
| MLP | BSLoRA (SS) | 46.20 | 57.42 | 79.95 | 83.33 | 81.88 | 73.95 | 81.83 | 48.52 | 69.14 | |
| | BSLoRA (GT) | 45.80 | 56.91 | 79.72 | 83.25 | 81.94 | 73.64 | 81.80 | 48.77 | 68.98 | 69.01 |
| | BSLoRA (KE) | 46.40 | 56.91 | 79.81 | 82.95 | 81.94 | 73.48 | 81.50 | 48.41 | 68.93 | |
| Attention & MLP | BSLoRA (SS) | 46.40 | 57.17 | 79.96 | 82.95 | 81.94 | 74.74 | 83.09 | 49.03 | 69.41 | |
| | BSLoRA (GT) | 46.20 | 56.83 | 79.89 | 82.87 | 81.94 | 74.27 | 82.97 | 48.36 | 69.17 | **69.25** |
| | BSLoRA (KE) | 46.40 | 56.67 | 80.04 | 83.08 | 82.15 | 73.64 | 82.60 | 48.98 | 69.18 | |

`Meta-Llama-3.1-8B`; Llama 1-13B: `https://huggingface.co/yahma/llama-13b-hf`; Qwen2.5-7B: `https://huggingface.co/Qwen/Qwen2.5-7B-Instruct`.

### B.2. Software and hardware configuration.

Our implementation utilizes the following configurations: *PyTorch* version 2.1.2, *Transformers* library version 4.41.0, *PEFT (Parameter-Efficient Fine-Tuning)* library version 0.11.1, *CUDA* version 12.4, *GPU:* NVIDIA V100 GPU with 32GB of memory, NVIDIA A100 GPU with 80GB, *Operating System:* Ubuntu.

### B.3. Datasets and Benchmarks

**BoolQ (Clark et al., 2019)** is a dataset for yes/no question answering. It consists of naturally occurring questions paired with passages extracted from Wikipedia. It is part of the SuperGLUE benchmark, a suite of challenging NLP tasks. It is used to assess a model's ability to perform reading comprehension and binary classification based on the context provided.

**PIQA (Bisk et al., 2020)** is a dataset designed to evaluate commonsense reasoning about physical interactions. It contains multiple-choice questions related to everyday physical tasks, asking models to choose the most plausible way of completing or describing an action. It is used as a benchmark for evaluating the commonsense reasoning abilities of language models, particularly in the context of tasks requiring physical understanding.

**HellaSwag (Zellers et al., 2019)** is a large-scale dataset for evaluating commonsense reasoning and natural language inference. The task involves selecting the most plausible continuation of a given story or event description from multiple choices. It is used to benchmark models on their ability to perform commonsense reasoning, particularly in cases where the correct answer requires understanding context, sequencing, and implications.

**WinoGrande (Sakaguchi et al., 2021)** is a large-scale dataset for commonsense reasoning, specifically designed to address the limitations of the Winograd Schema Challenge. The task involves resolving pronoun references in sentences, where the correct interpretation requires commonsense knowledge. It is used as a benchmark for evaluating models on their ability to perform pronoun resolution and commonsense reasoning.

**ARC-easy (Clark et al., 2018) and ARC-challenge (Clark et al., 2018)** are part of the AI2 Reasoning Challenge, designed to evaluate a model's ability to answer multiple-choice questions that require complex reasoning and background knowledge. They are used as a benchmark for testing advanced question-answering systems, especially those requiring sophisticated reasoning, knowledge integration, and inference capabilities.

**OpenbookQA (Mihaylov et al., 2018)** is a multiple-choice question-answering dataset that focuses on elementary science questions. The dataset comes with an "open book" of scientific facts, and models must combine this knowledge with reasoning to answer the questions correctly. It is used as a benchmark for evaluating a model's ability to perform open-domain question answering, where success requires not just knowledge retrieval but also reasoning and application of that knowledge.

**Social QA (Sap et al., 2019)** , often abbreviated as SIQA, is composed of question-answer pairs that simulate real-world information-seeking dialogues. This dataset is designed to assess the capability of models to engage in information-seeking

conversations, where the model must ask clarifying questions to a human user to gather information and then provide an answer to the original query.

**MMLU (Hendrycks et al., 2021)** is a benchmark designed to assess a model's world knowledge and problem-solving abilities across a wide range of subjects. It evaluates models in both zero-shot and few-shot settings, making the tasks more challenging and aligned with human evaluation methods. The benchmark spans 57 subjects, including STEM, humanities, social sciences, and other fields, with difficulty levels ranging from elementary to advanced professional. Each question presents four answer choices, and the task is to predict the correct one based on the given instruction.

