# OpenReview forum: "BSLoRA: Enhancing the Parameter Efficiency of LoRA with Intra-Layer and Inter-Layer Sharing"
_ICML.cc/2025/Conference — ICML 2025 poster_

### Official Review · Reviewer_qgmi · 2025-03-11

**Overall Recommendation:** 4

**Summary:**

This paper proposes Bi-Share LoRA (BSLoRA), a parameter-efficient fine-tuning approach for large language models (LLMs). The key idea is to improve upon standard Low-Rank Adaptation (LoRA) by introducing intra-layer and inter-layer parameter sharing to reduce the number of trainable parameters while maintaining or improving model performance.

**Claims And Evidence:**

The paper makes the following key claims, and most of them are supported by evidence in the experiments:
- Claim 1: BSLoRA reduces trainable parameters.
  - The results show that BSLoRA achieves around 50% parameter reduction while maintaining comparable or slightly improved performance over standard LoRA.
  - Limitation: The improvement is quite marginal, which raises concerns about whether the added complexity justifies the gains.
- Claim 2: The proposed shape transformation techniques enable flexible sharing.
  - The introduction of Kronecker Extension, Gate Transformation, and Slice Sharing provides different strategies for handling parameter shape mismatches, making parameter sharing feasible across different model components.

**Essential References Not Discussed:**

The paper covers the most relevant prior work.

**Experimental Designs Or Analyses:**

- The benchmarks (commonsense reasoning and MMLU) are widely used in LLM research and provide a fair basis for evaluating performance.
- The baselines (LoRA, VeRA, VB-LoRA, ShareLoRA, Tied-LoRA) are appropriate and competitive, ensuring that comparisons are meaningful.
- The absence of throughput, training speed, and memory usage comparisons in the main paper weakens the argument for parameter efficiency.

**Methods And Evaluation Criteria:**

The proposed BSLoRA method aligns with the general objective of improving parameter-efficient fine-tuning (PEFT) in LLMs by reducing trainable parameters while maintaining model performance.

**Other Comments Or Suggestions:**

No other suggestion.

**Other Strengths And Weaknesses:**

## Strengths
- The proposed multi-layer parameter-sharing strategy is well-motivated and provides clear parameter savings.
- The experiments are thorough, with evaluations of multiple models and tasks.
- Shape transformation techniques (Kronecker Extension, Gate Transformation) effectively solve parameter sharing constraints.

## Weaknesses
- Limited empirical improvements: The performance gain is marginal, making it unclear whether the complexity trade-off is justified.
- No throughput/memory efficiency evaluation: The paper claims efficiency improvements, but no evidence of speedup or memory savings is presented in the main paper.
- Not highly novel: The method mainly repackages known techniques (parameter sharing, Kronecker product, low-rank adaptation) rather than introducing fundamentally new ideas.
- Limited practical advantage: The main benefit only appears in large-scale multi-task settings (>200 tasks), which isn't a common real-world scenario.

**Questions For Authors:**

- How does BSLoRA impact training and inference speed?
- How does BSLoRA compare to LoRA in memory consumption during training?
- The performance gains are modest, questioning whether the increased architectural complexity is worth the trade-off.

**Relation To Broader Scientific Literature:**

The paper is related to prior work on parameter-efficient fine-tuning (PEFT):
- LoRA (Hu et al., 2022): The foundational method that BSLoRA extends.
- VeRA (Kopiczko et al., 2024) and VB-LoRA (Li et al., 2024): Parameter-sharing approaches that BSLoRA compares against.
- MultiLoRA (Wang et al., 2023) and HydraLoRA (Tian et al., 2024): Related work on adapting LoRA for multi-task learning.

the core techniques used (low-rank factorization, Kronecker product, and parameter sharing) are not new.

**Theoretical Claims:**

There are no deep theoretical contributions or proofs in this paper.

---

> ### Author Rebuttal · Authors · 2025-04-01
>
> **Comment:** "*Limited empirical improvements: The performance gain is marginal, making it unclear whether the complexity trade-off is justified.*"
>
> **Answer to C1**: Thank you for your comment. Please refer to R2C4.
>
> ---
>
> **Comment:** "*No throughput/memory efficiency evaluation: The paper claims efficiency improvements, but no evidence of speedup or memory savings is presented in the main paper.*"
>
> **Answer to C2**: Thank you for your comment. The goal of our method is to reduce parameter redundancy through effective parameter sharing. Our experimental results demonstrate that **BSLoRA achieves better performance with fewer trainable parameters**, indicating a significant improvement in **parameter efficiency**. In the appendix, we provide a detailed comparison of the **memory usage** between standard LoRA and BSLoRA under varying numbers of concurrently deployed downstream tasks. As shown in Figure 6, BSLoRA consumes **substantially less GPU memory during inference**, enabling **more downstream tasks to be deployed simultaneously**. This highlights the practical value of our method in **multi-task inference scenarios**, especially under memory-constrained conditions.
>
> We also test the inference speed comparison on the dataset of wikitext2 and ptb, the results are shown in https://anonymous.4open.science/r/BSLoRA-1E72/ . Results indicate that both GT and KE can achieve inference acceleration.
>
> ---
>
> **Comment:** "*Not highly novel: The method mainly repackages known techniques (parameter sharing, Kronecker product, low-rank adaptation) rather than introducing fundamentally new ideas.*"
>
>  **Answer to C3**: Thank you for the thoughtful question. As pre-trained language models continue to scale up, even parameter-efficient fine-tuning methods like LoRA introduce a substantial number of trainable parameters. Moreover, in real-world inference settings, LoRA adapters are typically kept separate from the backbone model to support simultaneous deployment of multiple downstream tasks. This makes reducing the memory footprint of LoRA adapters a critical challenge. Existing methods such as ShareLoRA focus only on **inter-layer parameter sharing**, neglecting the **intra-layer redundancy** that also contributes significantly to parameter inefficiency. **BSLoRA addresses this gap by introducing both intra-layer and inter-layer sharing mechanisms**, aiming to further reduce redundancy and achieve comparable or even improved performance with fewer trainable parameters. To overcome the challenge of sharing parameters with incompatible shapes, we introduce **three lightweight but effective shape transformation techniques** (SS, GT, KE), enabling flexible parameter sharing across diverse model components. While each individual technique may not be novel, their **systematic combination and targeted application to large language model fine-tuning** constitutes a meaningful contribution. Looking ahead, we plan to explore even more advanced sharing strategies to further enhance parameter efficiency.
>
> ---
>
> **Comment:** "*Limited practical advantage: The main benefit only appears in large-scale multi-task settings (>200 tasks), which isn't a common real-world scenario.*"
>
> **Answer to C4:** Reducing the size of LoRA adapters is practically important not only in large-scale multi-task settings (e.g., >200 tasks), but also increasingly so as model sizes continue to grow. For instance, applying LoRA with rank 64 to all linear layers in LLaMA 70B consumes approximately 1.4 GB of memory, and this rises to 4.5 GB for a 405B model. When deploying more than 10 tasks concurrently—common in real-world cloud services or multi-user systems—the cumulative memory usage of LoRA adapters alone can reach 45 GB, posing a significant burden for inference servers and edge devices alike. While smaller models may require hundreds of tasks to reach memory bottlenecks, larger models hit these limits even with far fewer concurrently deployed adapters. This makes parameter reduction a critical bottleneck for scaling both in terms of deployment cost and system responsiveness. BSLoRA addresses this challenge by reducing memory usage by over 50% compared to standard LoRA, enabling significantly more efficient multi-task deployment without sacrificing performance.

---

> > ### Comment · Reviewer_qgmi · 2025-04-09
> >
> > The authors' response has addressed my concerns regarding the efficiency of the method, so I have decided to raise my score.
> >
> >  I also noticed that some methods reduce model parameters through parameter sharing, such as Layerlink [1]. I suggest the authors consider citing this relevant work.
> >
> > [1]  "Layerlink: Bridging remote sensing object detection and large vision models with efficient fine-tuning." Pattern Recognition (2025): 111583.

---

> > > ### Author Response · Authors · 2025-04-09
> > >
> > > Thank you very much for your thoughtful follow‑up and for raising your score after reading our rebuttal. We truly appreciate the time and care you invested in evaluating our work. We are also grateful for your recommendation to reference Layerlink [1], which is highly relevant to our study. We agree that citing this work will strengthen the contextualization of our method. We will add the citation and discuss its relationship to our layer‑sharing strategy in the revised manuscript. Your constructive feedback has been invaluable in improving the quality and clarity of our paper. Thank you again for your support.

---

### Official Review · Reviewer_roJb · 2025-03-14

**Overall Recommendation:** 3

**Summary:**

The paper introduces Bi-Share LoRA (BSLoRA), which improves memory efficiency and inference speed by sharing parameters both within a layer (intra-layer) and across layers (inter-layer). The approach also introduces three shape transformation techniques—Slice Sharing, Gate Transformation, and Kronecker Extension—to ensure flexible parameter sharing across different module structures. Experimental results on LLaMA models (7B, 8B, and 13B) show that BSLoRA reduces parameters while improving or maintaining model performance on Commonsense Reasoning and MMLU benchmarks.

**Claims And Evidence:**

The claims made in the submission supported by clear and convincing evidence.

**Essential References Not Discussed:**

No

**Experimental Designs Or Analyses:**

- Even though the authors have conducted a contribution analysis by setting the rank of one sub-LoRA matrix to 8, the settings for Table 1 and Table 2 are not consistent. The ablation study evaluates the effectiveness of the three different modules by removing each component sequentially.

- Could you validate the effectiveness of BSLoRA by comparing it with ShareLoRA at different ranks, while keeping the number of trainable parameters similar?

**Methods And Evaluation Criteria:**

The paper claims reduced memory overhead, but does not provide empirical results on: Inference latency (Does BSLoRA speed up inference compared to LoRA?)

**Other Comments Or Suggestions:**

- Could you valid the effectiveness of BSLoRA by comparing with ShareLoRA with different ranks, under the case of similar trainable parameter?

**Other Strengths And Weaknesses:**

**Strengths:**

- The Bi-Share LoRA (BSLoRA) approach is interesting in how it combines intra-layer and inter-layer parameter sharing, addressing key limitations in LoRA-based fine-tuning.
- The method improves upon existing approaches such as VeRA, Tied-LoRA, and ShareLoRA, making it a significant contribution to parameter-efficient fine-tuning (PEFT).
- The paper is well-structured, with clear explanations of the methodology and mathematical formulations.

**Weaknesses:**

- The ablation study is relatively insufficient. See the section above, *"Experimental Designs or Analyses,"* and the section below, *"Questions for Authors,"* for details.

**Questions For Authors:**

- For the hyperparameter setting, how is r selected for different reshaping methods? Why not choose a consistent r across all methods?

- The Gate Transformation (GT) method yields a rank value equivalent to the local rank plus 2, likely due to one-rank gates causing some information loss. However, the performance of GT is comparable to the other two methods (i.e., SS and KE) in Table 1 and Table 2. How can this phenomenon be explained?

- “Specifically, the local component of LoRA performs best on HellaSwag, ARC-e, BoolQ, and SIQA, while the intra-layer shared component excels on PIQA and ARC-c.” Could you provide some insight into this phenomenon?

- “The results in Figure 4(c) show that modifying the constant improves both expressiveness and information content.” However, this conclusion is not obvious from Figure 4(c). How was it derived?

- In Table 10, different rank settings achieve similar performance, and the claim that 'Assigning a lower rank to the local component and a higher rank to the shared parameters yielded better performance, further illustrating the redundancy in the standard LoRA parameters' is not obvious.

**Relation To Broader Scientific Literature:**

The key contributions of the paper relate to prior work in parameter-efficient fine-tuning (PEFT), particularly in LoRA-based architectures. The paper builds on existing research in LoRA parameter sharing, addressing its limitations through a new bi-sharing approach that combines intra-layer and inter-layer weight sharing. While the paper contributes to efficiency, it would benefit from a more explicit connection to pruning or sparsity techniques in neural networks, as well as a deeper discussion on its applicability to broader PEFT techniques beyond LoRA .

**Theoretical Claims:**

No explicit mathematical proofs are presented, so there are no incorrect derivations to check.

---

> ### Author Rebuttal · Authors · 2025-04-01
>
> **Comment:** "*...  settings for Table 1 and Table 2 are not consistent. The ablation study evaluates the effectiveness of the three different modules by removing each component sequentially*."
>
> **Answer of C1**:  Thank you for your observation. The settings in Tab 1 and Tab 2 were designed to evaluate the **overall performance of BSLoRA in real-world scenarios**. Therefore, the rank settings in Tab 1 and Tab 2 were selected through extensive experiments to achieve **optimal performance** while maintaining parameter efficiency. In contrast, we chose rank=8 in the ablation study to clearly isolate each component’s contribution. These different configurations address different experimental goals.
>
> ---
>
> **Comment:** "*Could you validate the effectiveness of BSLoRA by comparing it with ShareLoRA at different ranks, while keeping the number of trainable parameters similar?*"
>
> **Answer to C2**: Thank you for your suggestion. We conducted additional experiments comparing **ShareLoRA at ranks 4, 8, 16, and 32** with **BSLoRA configured to have approximately same trainable parameters**. The results (https://anonymous.4open.science/r/BSLoRA-1E72/) show that at **lower budgets**, only the **KE** outperforms ShareLoRA. This suggests that under tight parameter budgets, **SS and GT have limited capacity to extract useful features**. However, as the rank increases, **all three variants consistently outperform ShareLoRA**, indicating that BSLoRA is more effective at leveraging **redundant capacity** when more parameters are available. These findings further support that there exists **significant redundancy both within and across layers**, and our sharing strategies can lead to **more efficient parameter utilization**, especially as the parameter budget increases.
>
> ---
>
> **Comment:** "*For the hyperparameter setting, how is r selected for different reshaping methods? Why not choose a consistent r across all methods?*"
>
> **Answer to C3**: Thank you for your insightful comment. SS, GT, and KE each have distinct structural characteristics, so they they naturally require different r configurations to achieve optimal performance. Using a single rank across all methods can be suboptimal, as each interacts with model layers differently. Instead, we determined **empirical best r values** through systematic experiments, balancing parameter efficiency and accuracy across benchmarks.
>
> ---
>
> **Comment:** "*... the performance of GT is comparable to the other two methods (i.e., SS and KE) in Table 1 and Table 2. How can this phenomenon be explained?*"
>
> **Answer to C4**: Your question is very interesting. One possible explanation is that GT **provides an implicit regularization effect, preventing overfitting to specific layers and encouraging a more generalizable representation**.  Additionally, the learned input and output transformations may help refine the information flow, ensuring that key features are preserved while removing redundant ones.
>
> **Comment:** '*“... local component of LoRA performs best on HellaSwag, ARC-e, BoolQ, and SIQA, while the intra-layer shared component excels on PIQA and ARC-c.” Could you provide some insight into this phenomenon?*'
>
> **Answer to C5**: Thank you for your insight. The varying effectiveness likely arises from differing task requirements. Tasks like PIQA and ARC-c may "share similar patterns", so consistent features from shared parameters help. Meanwhile, tasks like HellaSwag and BoolQ may "demand fine-grained contextual understanding", benefiting more from local parameter usage. This underscores BSLoRA’s hybrid design in adapting to different task needs.
>
> ---
>
> **Comment:** “*The results in Figure 4(c) show that modifying the constant improves both expressiveness and information content.” However, this conclusion is not obvious from Figure 4(c). How was it derived?*”
>
> **Answer to C6**: Thank you for your comment. The current radar plot in Figure 4(c) does not clearly show the improvement. We inferred it by comparing **average performance** across methods. The results used in this analysis are shown in the link above. We will clarify this in the revised paper and improve either the figure or the accompanying explanation.
>
> ---
>
> **Comment:** "*In Table 10, different rank settings achieve similar performance, and the claim that 'Assigning a lower rank to the local component and a higher rank to the shared parameters yielded better performance, further illustrating the redundancy in the standard LoRA parameters' is not obvious.*"
>
> **Answer to C7**: Thank you for your observation. Adjusting only the local component's rank merely brings minor performance improvement or worsen, indicating low-rank local component can achieve comparable performance. Besides, keeping a low-rank local component and adjusting shared components consistently results in comparable performance. Therefore, we can assign a lower rank to the local component and high ranks to shared components to achieve better parameter efficiency.

---

### Official Review · Reviewer_oFSX · 2025-03-16

**Overall Recommendation:** 2

**Summary:**

This paper introduce a method, sharing lora parameters across local, intra-layer, inter-layer. To address the shape mismatch issues, shape transformation are introduced including slice sharding, gate transformation, KRONECKEREXTENSION. Results on different datasets show the effectiveness of the method.

**Claims And Evidence:**

Yes.

**Essential References Not Discussed:**

No

**Experimental Designs Or Analyses:**

Partially.

**Concerns about Baseline Comparisons and Method Selection**

1. Fairness of Comparison with Baselines:
The comparison between BS LoRA and baselines such as ShareLoRA and Tied LoRA may be unfair. BS LoRA applies LoRA to both attention and MLP modules, whereas ShareLoRA and Tied LoRA are limited to attention modules only. This difference means BS LoRA updates more parameters, potentially inflating its performance relative to the baselines. To evaluate the true effectiveness of BS LoRA’s design, how does applying LoRA to both attention and MLP modules compare to applying it to only one module type (e.g., attention-only or MLP-only)? Does the combined approach yield synergistic benefits from updating both module types, or is the performance improvement merely a result of having more trainable parameters? Including ablation studies (e.g., separate versus joint tuning of attention and MLP modules) would help clarify the contribution of this design choice.

2. Rationale for Tied-LoRA Variant Selection:
The paper employs the TL5 variant of Tied-LoRA in its experiments, despite stating that the TL6 variant performs better. What motivated the choice of TL5 over TL6, and how does this decision influence the reported results? A clear explanation of this selection—or additional results using TL6—would provide greater confidence in the findings and ensure consistency between the claims and the experimental setup.

**Inconsistent Optimal Methods Across Setups**
Tab 2 results show that no single config among SS, GT, KE consistently outperforms others across tasks, model sizes, or ranks.

**The improvements compared with ShareLoRA and Tied LoRA is marginal.**

**Methods And Evaluation Criteria:**

Yes.

**Other Comments Or Suggestions:**

No.

**Other Strengths And Weaknesses:**

While the paper has some weaknesses, as discussed above, it also demonstrates notable strengths that enhance its value. The writing is clear and accessible, effectively simplifying the complex topic of shape mismatching in machine learning models. Additionally, the paper proposes multiple innovative solutions to address shape mismatching, each accompanied by a balanced discussion of its advantages and disadvantages. This thorough approach highlights the authors' expertise and offers practical insights for addressing similar challenges.

**Questions For Authors:**

Please see the above questions.

**Relation To Broader Scientific Literature:**

The key contributions of BS LoRA are deeply tied to the broader scientific literature on parameter-efficient fine-tuning of LLMs. It builds on the foundational work of LoRA and extends the parameter-sharing ideas pioneered by ShareLoRA and Tied-LoRA. By introducing a block-wise sharing strategy, BS LoRA offers a new approach that seeks to improve upon prior findings—balancing the reduction of trainable parameters with the need for layer-specific adaptability. This work contributes to the evolving field of PEFT, where efficiency remains a critical concern for deploying LLMs in resource-constrained environments.

**Theoretical Claims:**

No proof.

---

> ### Author Rebuttal · Authors · 2025-04-01
>
> **Comment:** "*Fairness of Comparison with Baselines: The comparison between BSLoRA and baselines such as ShareLoRA and Tied LoRA may be unfair. ... To evaluate the true effectiveness of BS LoRA’s design, how does applying LoRA to both attention and MLP modules compare to applying it to only one module type (e.g., attention-only or MLP-only)? ....*"
>
> **Answer to C1:** Thank you for the thoughtful suggestion. We agree that evaluating the contribution of parameter sharing across different module types is important for a fair comparison. To address this, we conducted an additional ablation study, comparing BSLoRA applied to **only the attention modules**, **only the MLP modules**, and **both modules simultaneously**.
>
> The results show that applying BSLoRA to the **attention modules alone yields better performance** than applying it only to the MLP modules, indicating that **redundancy in attention layers is more significant**. Moreover, applying BSLoRA to **both attention and MLP modules further improves performance** over attention-only tuning. This suggests that **redundancy exists not only across layers but also across different module types within the same layer**, as discussed in Section 2.2. These findings support our core claim: combining **intra-layer and inter-layer sharing** strategies enables more effective reduction of parameter redundancy, while also contributing to better overall performance. The results are shown in https://anonymous.4open.science/r/BSLoRA-1E72/ .
>
> ---
>
> **Comment:** "*Rationale for Tied-LoRA Variant Selection*"
>
> **Answer to C2:** Thank you for raising this important question. After reviewing the original Tied-LoRA paper, we observed that **both TL5 and TL6 perform well** across various tasks. In our own experiments, we replicated both the TL5 and TL6 configurations. Based on the specific training setup and evaluation methods used in our study, **we found that TL5 outperformed TL6 under these conditions**. Consequently, we decided to present the results for TL5 in the main text of the paper. We will clarify this choice in the revised version of the paper and provide a detailed comparison between TL5 and TL6 in the appendix to ensure transparency and consistency in the reported findings. The results are shown in https://anonymous.4open.science/r/BSLoRA-1E72/ .
>
> **Inconsistent Optimal Methods Across Setups** Tab 2 results show that no single config among SS, GT, KE consistently outperforms others across tasks, model sizes, or ranks.
>
> **Answer to C3 :**  Thank you for your thoughtful observation. Indeed, the optimal configuration of **SS**, **GT**, and **KE** varies across tasks, model sizes, and ranks. This variability arises due to the **task-specific nature of the benchmarks** and the **differences in model layer architectures**. Our approach aims to provide **flexibility** by offering multiple methods that can be tailored to the requirements of different tasks and model sizes.
>
> For example, as shown in **Table 2**, the **SS method performs best** on **Llama 1-7B**, while the **KE method performs best on Llama 3-8B**. This flexibility allows users to select the method that fits their deployment context. Moreover, the performance variation across different methods is minimal, highlighting the **robustness of BSLoRA** across diverse settings.
> We believe this **flexibility** is one of BSLoRA’s strengths, enabling it to be adapted for various deployment scenarios without the need for extensive hyperparameter tuning for each task.
>
> ---
>
> **Comment:** "*The improvements compared with ShareLoRA and Tied LoRA are marginal.*"
>
> **Answer to C4:** Thank you for your comment. While the performance gains of BSLoRA over other parameter-sharing methods such as ShareLoRA and Tied-LoRA may appear modest, our method introduces a **more fine-grained parameter sharing strategy**. As discussed in Section 2.2 of our paper, redundancy in LoRA parameters exists not only across layers but also **within the same layer across different parameter modules**. However, existing approaches focus only on inter-layer sharing for modules at the same relative position, **overlooking the redundancy among intra-layer components**.
>
> BSLoRA addresses this limitation by jointly implementing **intra-layer and inter-layer parameter sharing**, enabling a more comprehensive reduction in trainable parameters. To overcome the challenge of sharing across weights with mismatched shapes, we further propose **three shape transformation methods** (SS, GT, KE), which allow the model to flexibly align and share information across diverse modules. This design increases the **parameter efficiency** and enhances the model’s ability to capture shared features across tasks. Our experimental results show that BSLoRA not only achieves a further reduction in parameter redundancy but also delivers **consistent performance improvements**, showing a better balance between efficiency and effectiveness than existing sharing-based methods.

---

### Official Review · Reviewer_BXVq · 2025-03-17

**Overall Recommendation:** 3

**Summary:**

In this paper, the authors proposed BSLoRA that add intro-layer and inter-layer sharing for LoRA to reduce trainable parameters while maintenance the performance. Multiple experiments were conducted on several benchmark datasets and showed slightly better performance.

**Claims And Evidence:**

Yes.

**Essential References Not Discussed:**

No, essential references are adequate.

**Experimental Designs Or Analyses:**

Yes, the main experimental results, analysis, and ablation study.

**Methods And Evaluation Criteria:**

Yes.

**Other Comments Or Suggestions:**

1. Typo in the caption of Figure 1: "(left)" and "(right)" should be "(top)" and "(bottom)".

**Other Strengths And Weaknesses:**

***Strengths***
1. The idea of adding intra-layer and inter-layer sharing to further reduce trainable parameters of LoRA makes sense and is technically sound to me.
2. The authors conducted extensive on experiments, analysis, and ablation study to validate the proposed approach.
3. Writing is good and easy to follow.

*** Weaknesses***
1. Given that LoRA already reduced the trainable parameters dramatically compared to full training, the additional reduction from the proposed intra-layer and inter-layer sharing is relatively small.
2. Given the complexity of hyper-parameter tuning and shape transformation (SS, GT, KE), I am not fully convinced that the gain would outweigh the effort.

**Questions For Authors:**

Please refer to the "Other Strengths And Weaknesses" for the detailed comments and provide additional justification regarding the balance between benefits gained and required effort.

**Relation To Broader Scientific Literature:**

The introduce intra-layer and inter-layer sharing approach further reduced the trainable parameters of LoRA and could enable fine-tuning of much larger models.

**Theoretical Claims:**

No theoretical claims.

---

> ### Author Rebuttal · Authors · 2025-03-31
>
> **Comment:** "*Given that LoRA already reduced the trainable parameters dramatically compared to full training, the additional reduction from the proposed intra-layer and inter-layer sharing is relatively small.*"
>
> **Answer to C1**: Thank you for your insightful comment. While it is true that LoRA significantly reduces trainable parameters compared to full fine-tuning, the memory footprint of LoRA adapters becomes increasingly non-negligible as the size of the pre-trained model and the number of downstream tasks grow. In real-world inference systems, **LoRA adapters are typically not merged with the base model** to support dynamic multi-task serving. For instance, applying LoRA with rank 64 on LLaMA 70B introduces over 1.4 GB of additional memory. As more tasks are deployed simultaneously, e.g., **100 tasks would take about 140G memory, this overhead quickly becomes a major bottleneck.**
>
> To address this, recent works such as Vera, ShareLoRA, and Tied-LoRA explore further parameter reduction. Building on these efforts, BSLoRA introduces **a more fine-grained parameter sharing** framework that combines local, intra-layer, and inter-layer components. To enable sharing across weights of different shapes, we also propose three shape transformation strategies (SS, GT, KE). This design not only reduces trainable parameters but also improves model performance, as demonstrated in our experiments. From our experiment's results, **BALoRA can reduce over 50% trainable parameters on average and achieve an average 1.25% performance improvement.** Importantly, BSLoRA can be seen as **a flexible superset of LoRA**, offering a tunable architecture that adapts to both performance and efficiency needs.
>
> ---
>
> **Comment:** "*Given the complexity of hyper-parameter tuning and shape transformation (SS, GT, KE), I am not fully convinced that the gain would outweigh the effort.*"
>
> **Answer to C2**: We appreciate your concerns regarding the complexity of hyperparameter tuning and shape transformation. In practice, **BSLoRA does not introduce additional tuning overhead compared to standard LoRA**—the only tuning required is selecting appropriate rank values for the chosen sharing strategy. Importantly, **all three proposed sharing strategies (SS, GT, KE) consistently reduce parameter count while maintaining or improving performance**, making the selection process flexible and low-effort. In our experiments, we empirically chose rank settings that best fit each sharing method, but our **ablation studies (Tables 7, 8, and 9) show that BSLoRA performs robustly across different configurations**, demonstrating that extensive hyperparameter tuning is not necessary for effective deployment. By default, we adopt the **Kronecker Extension (KE)** strategy in our setup, as it offers the best trade-off between **parameter efficiency and performance**. Overall, BSLoRA is a simple yet effective solution that achieves higher efficiency **without adding tuning complexity**.
>
> ---
> **Comment**: "*Typo in the caption of Figure 1: (left) and (right) should be (top) and (bottom)*"
>
> **Answer to C3**: Thank you for pointing this out. We will correct the caption of Figure 1 from "(left)" and "(right)" to "(top)" and "(bottom)" in the revised version.

---

> > ### Comment · Reviewer_BXVq · 2025-04-08
> >
> > Really appreciate the authors' rebuttal, increasing my rating to weak accept now.

---

> > > ### Author Response · Authors · 2025-04-09
> > >
> > > Thank you very much for your thoughtful feedback. We sincerely appreciate your insightful suggestions, which have contributed significantly to improving our work. Your support is invaluable, and we are grateful for the time and effort you have put into reviewing our submission.

---

### Decision · Program_Chairs · 2025-05-01

**Decision:**

Accept (poster)

**Comment:**

The paper introduces BSLoRA, a new variant of LoRA fine-tuning aimed at reducing the overall number of adapter parameters while preserving/improving performance.  The approach of weight tying to reduce LoRA parameters is not new, yet BSLoRA innovates on previous work through two major contributions: i) a decomposition of adapters into (Local, Intra, Inter)-layer tied parameters--Local is at the vanilla LoRA level, Intra is at the Transformer-layer level, and Inter is at the global level--and (ii) three shape transformation techniques to deal with mismatched tensor dimensions in the Intra- and Inter-layer adapters.  By reducing the dimension of Local-layer adapters while retaining larger dimensions for the Intra- and Inter-layer adapters, BSLoRA largely outperforms related parameter-tying LoRA variants (VeRA, VB-LoRA, ShareLoRA, and Tied-LoRA) for Llama-3 and Llama-1 models across the widely used common sense reasoning and MMLU (0- and 5-shot) benchmarks.  The authors perform additional experiments demonstrating the effective rank of BSLoRA, the practicality memory savings using BSLoRA for massive-scale LoRA adapter service (e.g., Multi-LoRA), and ablations for the various shape transformation methods (evaluated on GSM8K and MMLU).

The paper is well written, the method is an intuitive improvement over existing methods, and the experiments extensively demonstrate the soundness of the approach.  The majority of reviewers agree that the work is worthy of acceptance.  Reviewers BXVq and qgmi both raised concerns of the need for reducing LoRA memory overhead, but the authors addressed these concerns, resulting in qgmi raising their score.  While I would have liked to see other model families evaluated to explore the effectiveness of shape transformations and other fine-tuning datasets than the Alpaca dataset utilized, the overall experiments are comprehensive and convincing.  Reviewer oFSX was in favor of weak reject, but did not participate in either the AC-reviewer or author discussions throughout the review process.

For the camera ready, I ask the authors to include a discussion in how their method can be integrated with S-LoRA in future work and extensively highlight the MultiLoRA results in the introduction to motivate the practical need for BSLoRA.